# Reliable Active Learning via Influence Functions

**Meng Xia**                                                                    *meng.xia@duke.edu*
*Department of Electrical & Computer Engineering*
*University of Duke*

**Ricardo Henao**                                                         *ricardo.henao@duke.edu*
*Department of Electrical & Computer Engineering*
*University of Duke*
*Biological, Environmental Sciences and Engineering Division*
*King Abdullah University of Science and Technology (KAUST)*

**Reviewed on OpenReview:** *https: // openreview. net/ forum? id= dN9YICB6hN*

## Abstract

Due to the high cost and time-consuming nature of collecting labeled data, having insufficient labeled data is a common challenge that can negatively impact the performance of deep learning models when applied to real-world applications. Active learning (AL) aims to reduce the cost and time required for obtaining labeled data by selecting valuable samples during model training. However, recent works have pointed out the performance unreliability of existing AL algorithms for deep learning (DL) architectures under different scenarios, which manifests as their performance being comparable (or worse) to that of basic random selection. This behavior compromises the applicability of these approaches. We address this problem by proposing a theoretically motivated AL framework for DL architectures. We demonstrate that the most valuable samples for the model are those that, unsurprisingly, improve its performance on the entire dataset, most of which is unlabeled, and present a framework to efficiently estimate such performance (or loss) via influence functions, pseudo labels and diversity selection. Experimental results show that the proposed *reliable active learning via influence functions* (RALIF) can consistently outperform the random selection baseline as well as other existing and state-of-the art active learning approaches.

## 1 Introduction

In the last decade, deep learning (DL) models have achieved record-breaking performance on datasets with millions or even billions of samples (Chen et al., 2022; 2023; Jain et al., 2022). However, their broad practical applicability in scenarios where data is scarce has been limited mainly due to the lack of sufficient (and high-quality) labeled data for training. This is often due to the high cost or time-consuming nature of manual labeling (or sample annotation) often requiring subject-matter expertise. Aiming to address this gap in the field, active learning (AL), which has been historically applied to traditional machine learning approaches, emerged as a means to improve the applicability of deep learning models in such scenarios. The main objective of AL is to design strategies to selectively annotate (label) small subsets of a large (unlabeled) dataset so that model performance can be maximized while controlling the cost associated with annotations (Settles, 2009; Sener & Savarese, 2017). It is worth noting that there are also situations where both data (*e.g.*, images) and labels are scarce, however, such scenario is outside of the scope of this work.

Early AL algorithms (Lewis & Gale, 1994; Dagan & Engelson, 1995; McCallum et al., 1998; Tong & Koller, 2001; Freytag et al., 2014) were originally developed for traditional machine learning models, such as logistic regression (Kleinbaum et al., 2002), support vector machines (SVMs) (Hearst et al., 1998), or naive Bayes classifiers (Rish et al., 2001). Unfortunately, applying these algorithms directly to deep neural networks is challenging because they rely on unique characteristics of traditional machine learning models (*e.g.*, the

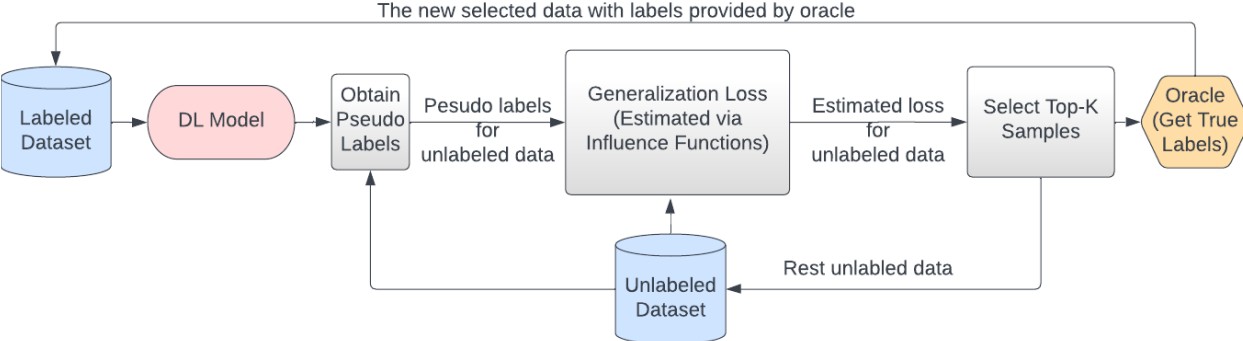

Figure 1: The basic reliable active learning via influence functions (RALIF) framework. Starting with a small labeled dataset and a much larger unlabeled dataset, at each selection cycle, we first train the target DL model with the labeled dataset and generate pseudo labels for the entire unlabeled dataset using the trained target DL model. We then use the selection mechanism based on influence functions to rank the most likely useful samples from the unlabeled dataset, query the oracle to obtain the true labels for the top $K$ ranked samples, and update the labeled and unlabeled datasets accordingly. This process is repeated for a fixed number of cycles or until the labeling budget has been spent or performance target has been achieved.

convexity of their learning objective), and the widely known, but incompatible, need for large datasets when training DL models. As an example of the former, Tong & Koller (2001) leveraged the *version space duality* in SVMs consisting in that each training sample corresponds to a hyperplane in the version (solution) space of SVMs (Vapnik, 1999; Herbrich et al., 2001), and aimed to select and label samples that have the most significant impact on shrinking such solution space, which is a sensible approach considering that SVMs are known to have a *unique* optimum classification hyperplane. Unfortunately, there is no theoretical basis or direct evidence to support the assumption that each sample in a deep neural network corresponds to a hyperplane in their solution space. In fact, neural networks have in general non-convex optimization objectives known for not yielding a unique global optimum, and instead being overwhelmed by local optima.

In contrast, numerous DL-based AL algorithms have been proposed specifically for deep learning models (Gal et al., 2017; Beluch et al., 2018; Sinha et al., 2019; Yoo & Kweon, 2019; Liu et al., 2021; Wang et al., 2022; Yi et al., 2022; Vo et al., 2022). While DL-based AL algorithms have demonstrated strong performance in some scenarios, recent work by Ash et al. (2019); Munjal et al. (2022); Wang et al. (2022) revealed the performance unreliability of many DL-based AL methods across different scenarios. Specifically and quite often, existing DL-based AL algorithms (Wang & Shang, 2014; Sener & Savarese, 2017; Gal et al., 2017; Sinha et al., 2019) fail to consistently outperform the most basic (no skill) random AL selection mechanism, which assumes that all (unlabeled) samples are equally likely to improve the performance of the model if labeled and added to the training set. This unreliability undermines their applicability and adoption, particularly in situations when labeling costs are high.

Motivated by such findings, we propose a more reliable AL algorithm for DL architectures we call *reliable active learning via influence functions* (RALIF), which benefits from a theoretically grounded selection mechanism that increases the likelihood of selecting samples that will yield performance improvements when labeled and included in dataset used for model training. The RALIF framework illustrated in Figure 1 introduces a new objective for AL that leverages the concept of *influence function* (Hampel, 1974) to make the sample selection process more efficient. Specifically, Koh & Liang (2017) showed that influence functions can approximate the changes in model parameters caused by incorporating new samples into the training dataset, but without the need for explicitly retraining the deep neural network. This enables RALIF, in principle, to efficiently quantify the generalization ability of a model by estimating the influence of new samples on the complete-dataset performance, thus enabling informed decisions during the AL sample selection process. Our contribution consists on leveraging pseudo labels to repurpose these influence functions for the AL case where labels are not available for most of the dataset. Experimental results on three datasets (CIFAR10, FICAR100 and iNaturalist) demonstrate that RALIF achieves consistent and superior performance compared to random

selection (as a no-skill baseline), as well as existing AL approaches across various practical scenarios, thus providing strong empirical evidence of the reliability and effectiveness of RALIF.

The rest of this paper is organized as follows, in Section 2 we discuss the related work in the context of the RALIF framework. In Section 3, we provide a detailed description of the notation used in this paper and define the standard setting for a (sequential) AL algorithms. In Section 4, we introduce the proposed AL objective and describe the strategies we employ to implement RALIF with deep learning neural networks. Subsequently, in Section 5, we present the results of our experiments, which highlight the performance reliability of the proposed AL framework across different scenarios, and conclude in Section 6 with a discussion and directions for future work.

## 2 Related Work

**AL for traditional machine learning models**   Among the early AL algorithms, *uncertainty* sampling stands out as one of the most commonly used. This strategy aims to selecting samples for which predictions with the model trained on the current labeled dataset exhibit the highest uncertainty. The rationale behind this approach is that selecting samples with uncertain predictions is believed to provide additional information to the model that can likely improve its performance relative to the current version of the trained model. For example, Lewis & Gale (1994) introduced a probabilistic binary classifier that utilized a bag-of-words representation for text data and proposed selecting samples with posterior predictive probabilities close to the decision boundary. In another example involving multi-class text classification tasks, Settles & Craven (2008) suggested selecting samples for which the entropy of the predicted outputs of a conditional random field (Lafferty et al., 2001) is high, as a proxy for uncertainty.

Another popular early introduced AL strategy is *query-by-committee* (Seung et al., 1992). This approach consists of constructing a committee of models trained on a labeled dataset. The selection process typically involves choosing samples that elicit the most diverse or heterogeneous predictions among the committee members. Importantly, constructing the committee is a key aspect of this approach. For instance, McCallum et al. (1998) constructed the committee by training a naive Bayes classifier with a labeled text dataset and sampled multiple naive Bayes classifier instances from the posterior distribution of the trained parameters. Alternatively, Dagan & Engelson (1995) trained a hidden Markov model (HMM) (Rabiner, 1989) with a labeled text dataset and constructed the committee by sampling a collection of HMMs based on the posterior distribution of the trained HMM parameters. Moreover, there are several other early AL strategies, such as *Variance Reduction* (Cohn et al., 1996) or *Fisher Information Ratio* (Zhang & Oles, 2000), which aim to minimize the expected generalization error of the model by selecting unlabeled samples that minimize the variance or the Fisher information ratio estimated for the model including such samples. See Settles (2009) for a comprehensive overview of these methods.

**AL for deep neural networks**   Over the past few years, an abundance of DL-based AL algorithms have been proposed, most of which can be categorized into three main strategies, namely, *i*) based on uncertainty (Wang & Shang, 2014; Beluch et al., 2018), *ii*) based on diversity (Sener & Savarese, 2017; Geifman & El-Yaniv, 2017), and *iii*) hybrid approaches (Lughofer, 2012; Ash et al., 2019; Wu et al., 2021).

Uncertainty-based AL algorithms for DL architectures built upon the ideas of the early uncertainty sampling strategies for AL (Hwa, 2004; Settles & Craven, 2008), where uncertainty is typically estimated by calculating the entropy of the model predictions for each unlabeled sample (Wang & Shang, 2014). In addition to directly using the model's predictions to estimate uncertainty, several other methods have been proposed to estimate the uncertainty of deep neural networks for unlabeled samples. One of such methods, proposed by Gal & Ghahramani (2016), involved applying dropout (Srivastava et al., 2014) before each layer of the neural network. By sampling from the dropout masks, multiple predictions could be obtained for each unlabeled sample. These predictions were then averaged to obtain the final prediction for the sample, and the entropy of the *expected* prediction could be calculated as a measure of uncertainty. Another approach, presented by Beluch et al. (2018), followed a similar framework to Gal & Ghahramani (2016), but used a different method to obtain multiple predictions for unlabeled samples. Instead of using dropout, Beluch et al. (2018) trained multiple models with identical architecture but different random initialization seeds using the same

labeled dataset. By obtaining predictions from these independently trained models, multiple predictions for unlabeled samples could be obtained, which could then be used to estimate uncertainty. Note that this approach is more computationally expensive than simply using dropout like in Gal & Ghahramani (2016).

In diversity-based AL algorithms, samples that are representative of the entire dataset distribution are preferred. For instance, in Sener & Savarese (2017), features extracted from the model trained on the labeled dataset were used as representations for the unlabeled samples. Then, their goal was to select a subset (batch) of samples that minimized the maximum distance of any arbitrary sample from the dataset to its nearest selected sample. Similarly, Geifman & El-Yaniv (2017) also used features extracted from the model trained on the labeled dataset as representations for the unlabeled samples. They focused on selecting samples that were farthest away from the labeled dataset. By selecting samples that were farthest from the labeled dataset, they aimed to capture the most diverse and informative samples that are distinct from those already labeled.

Hybrid strategies typically combine uncertainty and diversity approaches to select samples that are both uncertain and highly representative of the entire unlabeled dataset. For instance, Lughofer (2012) proposed a two-stage hybrid active learning algorithm. In the first stage, they employed the evolving vector quantization approach (Lughofer, 2008) to cluster the data. In the second stage, they selected samples that were either close to the cluster centers (highly representative) or near the boundaries of clusters (highly uncertain). In Ash et al. (2019), a similar approach to the diversity strategy presented in Sener & Savarese (2017) was employed, but with a consideration of the trade-off between uncertainty and diversity. Pseudo labels were generated for the unlabeled samples using the model trained on the labeled dataset, and the gradients of the loss function with respect to the model parameters were computed. The gradients of the unlabeled samples were then used as representations for a clustering algorithm. The first cluster center was selected as the unlabeled sample with the largest norm of gradients (uncertainty), and subsequent centers were selected based on their distance to the nearest center (diversity). Another study by Wu et al. (2021) introduced a hybrid active learning approach using two deep neural networks. One network was trained to estimate the diversity of the unlabeled samples, while the other network estimated the loss for each unlabeled sample. Samples with high diversity (representativeness) and high loss (uncertainty) were then selected for labeling.

There are also AL algorithms specifically designed based on the understanding or techniques specific to deep neural networks. For example, Sinha et al. (2019) leveraged adversarial training to select unlabeled samples that were most dissimilar from labeled samples. Yoo & Kweon (2019) posited that samples with the largest loss indicated the largest informativeness, thus they trained a module to estimate loss for unlabeled samples, then selected samples with the largest predicted loss. Furthermore, in their study, Liu et al. (2021) aimed to select samples capable of reducing the loss of a labeled reference set. More recently, Wang et al. (2022) proved that samples that mostly change the DNN weights were most useful, and accordingly, they selected samples that generated the largest approximate gradients.

It is important to highlight that while our work shares similarities with some existing AL algorithms for DNNs, there significant differences exist between our approach and others. For example, though both the approaches introduced by Ash et al. (2019) and Wang et al. (2022), as well as the proposed RALIF, involve the utilization of sample gradients, our motivations for sample selection differ significantly. In Ash et al. (2019), the authors argue that samples for which the current model produce uncertain predictions will yield larger gradients. Thus, they prefer to select samples with larger gradients. On the other hand, Wang et al. (2022) adopts a different perspective. They propose that the magnitude of gradients introduced by samples, *i.e.,* the extent to which samples influence the DNN weights, can serve as an upper bound for the loss within unobserved test data. As such, they consider that lowering the amount of samples with large gradients will result in a higher upper bound for the loss of the unobserved test dataset, implying they too favor selecting samples with larger gradient magnitudes. In contrast, the proposed RALIF leverages sample gradients and influence functions to assess the impact of samples on the entire dataset. To further enhance the estimation of the influence of unlabeled samples on the entire dataset, we introduce the use of using pseudo labels to approximate the true dataset distribution.

When comparing our approach to Liu et al. (2021), the most significant distinction lies in the objective they use to select samples that reduce the task loss on a labeled *reference* set, whereas our approach prioritizes

selecting samples that reduce the task loss of the entire dataset. It is crucial to note that their algorithm (ISAL: influence selection for active learning) necessitates a labeled reference set in addition to the initial labeled dataset, resulting in the need for annotations not only for the initial labeled dataset but also for this additional reference set at the outset of the active learning selection process. In contrast, RALIF only requires annotations for the initial labeled dataset, thus underscoring that in principle, we need fewer annotations overall. Instead of constructing a reference set with labels, we rely on pseudo-labels to effectively approximate the true dataset distributions, and we provide theoretical results in the following sections to substantiate this approximation. Another notable difference is the incorporation of a diversity selection operation in our methodology. As demonstrated by the results in Table 1, this diversity operation ensures the selection of samples that are not only valuable but also diverse, resulting in an enhanced overall performance for our approach. We also provide experimental comparisons between RALIF and their approach under a specific active learning setting in Appendix A.6. The results indicate the superior performance of RALIF compared to their proposed ISAL method.

**Reliability in active learning**   While most AL algorithms for traditional machine learning models with convex objectives enjoy theoretical guarantees for their performance (Tong & Koller, 2001; Brinker, 2003; He et al., 2004), this is in general not the case for DL-based AL algorithms. For instance, uncertainty- and DL-based AL algorithms (Wang & Shang, 2014; Gal et al., 2017) have been shown to perform comparably to basic random selection in several scenarios (Sener & Savarese, 2017), whereas the performance of Wang & Shang (2014) was even slightly worse than random selection in the early stages of the AL selection process (Ash et al., 2019). Studies by Yoo & Kweon (2019) and Wang et al. (2022) revealed that DL-based AL algorithms with theoretical support, such as Sener & Savarese (2017) and Gal et al. (2017), also failed to consistently outperform random selection. Further, experiments conducted by Munjal et al. (2022) demonstrated that many DL-based AL algorithms, including Wang & Shang (2014); Sener & Savarese (2017); Gal et al. (2017); Sinha et al. (2019), exhibited inconsistent performance when compared to random selection under the same experimental conditions. In their experiments, the conditions referred to applying these algorithms on the same datasets, using the same classification model, and employing the same sample selection size at each active learning selection cycle; all of this to make comparisons easier and fairer.

This reported inconsistency in performance undermines the applicability and adoption of AL algorithms in real-world scenarios. Therefore, we are motivated to propose a DL-based AL algorithm that is not only theoretically grounded, but also demonstrates reliable performance across various AL selection scenarios, *e.g.*, different datasets, model architectures and selected sample size (batch size) in each AL selection cycle.

## 3   Problem Definition (Active Learning)

We consider a hypothetical multi-class classification dataset $D = \{(x_i, y_i)\}_{i=1}^n$, where $n$ is the total number of samples, $x_i \in \mathcal{X}$ represents an input sample (*e.g.*, images in our experiments), and $y_i \in \mathcal{Y} = \{1, ...K\}$ represents its corresponding class label, which is not available for the majority of the dataset $D$. In active learning, we begin with a small labeled subset $L_0 = \{(x_i, y_i)\}_{i=1}^{m_0}$ containing $m_0 \ll n$ labeled samples. The remaining samples, which are unlabeled, are denoted as the set $U_0 = \{x_j\}_{j=m_0+1}^n$. Importantly, in active learning (AL), we do not have access to the true labels of the samples in $U_0$, denoted as $U_0^Y = \{y_j | x_j \sim U_0\}_{j=m_0+1}^n$. Let $M$ denote a classification learner, *e.g.*, a deep neural network, and let $M_{L_0}$ denote the learner $M$ trained with labeled dataset $L_0$. Further, $l(\cdot, \cdot; M) : \mathcal{X} \times \mathcal{Y} \to \mathbb{R}$ denotes the loss function used during the training process, *e.g.*, the standard cross-entropy loss for classification tasks.

In each AL selection cycle $c$ (full model training iteration), a sequential AL algorithm selects a single sample $x_s \sim U_{c-1}$ from the unlabeled set $U_{c-1}$ at cycle $c-1$, and queries its true label $y_s$ with an oracle function[1], resulting in a new labeled dataset $L_c = L_{c-1} \cup (x_s, y_s)$. The most basic (*no skill*) strategy for AL is random selection, where a sample $x_s$ is randomly chosen from $U_{c-1}$ and its true label is queried; this is denoted as the *Random* strategy. Arguably the most common baseline strategy is the *Uncertainty* strategy, which selects the sample $x_s$ from $U_{c-1}$ with the largest predictive uncertainty under the current model $M_{L_{c-1}}$.

---

[1]In practice, this is replaced by a human or expert observing the sample and producing the label of interest.

Such uncertainty is quantified in terms of the entropy (Shannon, 1948). A more detailed information about these two strategies can be found in Section 5.

# 4 Reliable Sample Selection for Active Learning

We propose a selection strategy that aims to pick samples from the unlabeled set $U_{c-1}$, denoted as $x_s \sim U_{c-1}$, such that the loss of $M_{L_c}$ on the entire set $D$ is minimized. This objective reflects the goal of improving the generalization performance of $M_{L_c}$ on the entire dataset $D$, which in turn is a proxy for its generalization ability. Formally then, the objective of the sequential AL algorithm in each cycle $c$ can be formulated as

$$\underset{x_s \sim U_{c-1}}{\arg\min} \ \mathbb{E}_{(x_i,y_i)\sim D}[l(x_i,y_i;M_{L_c})] = \frac{1}{n}\sum_{i=1}^{n}l(x_i,y_i;M_{L_c}). \tag{1}$$

There are two key barriers preventing us from using (1) directly, namely, $i$) at cycle $c$ we do not have access to the label for $x_s$ to be able to obtain $M_{L_c}$ via labeled set $L_c = L_{c-1} \cup (x_s, y_s)$, and $ii$) we do not have access to most of the labels in $D$, but only to those in $L_{c-1}$.

Below, we describe how we propose approximating (1) to use it as a means to select samples that are likely to improve the generalization performance of the model $M_{L_c}$ on $D$. Specifically, we will introduce a variant objective based on (1) as a modification to the original AL objective, which underscores the issue concerning lacking most of the labels in $D$. Then, we will propose strategies to address the practical implementation challenges that arise when applying such an objective variant in practice. More precisely, we will address the challenges associated with lacking labels for most samples and the expensive selection process resulting due to the time-consuming nature of DL model training. Finally, we motivate the need for selecting multiple samples (a batch) in each cycle in the context of deep learning. Taken together, the objective, implementation strategies and batch selection constitute the proposed RALIF which is conceptually summarized in Figure 1.

## 4.1 AL Objective Revisited

We can readily derive the following upper bound for (1):

$$\mathbb{E}_{(x_i,y_i)\sim D}[l(x_i,y_i;M_{L_c})] = \frac{1}{n}\sum_{i=m_c+1}^{n}l(x_i,y_i;M_{L_c}) + \frac{1}{n}\sum_{i=1}^{m_c}l(x_i,y_i;M_{L_c})$$

$$\leq \frac{1}{n-m_c}\sum_{i=m_c+1}^{n}l(x_i,y_i;M_{L_c}) + \frac{1}{m_c}\sum_{i=1}^{m_c}l(x_i,y_i;M_{L_c}) \tag{2}$$

$$= \underbrace{\mathbb{E}_{(x_i \sim U_c, y_i \sim U_c^Y)}[l(x_i,y_i;M_{L_c})]}_{\text{Unlabeled subset loss}} + \underbrace{\mathbb{E}_{(x_i,y_i)\sim L_c}[l(x_i,y_i;M_{L_c})]}_{\text{Training loss}},$$

in which we made explicit that the labels for $U_c$ denoted as $U_c^Y$ are not available in practice. From (2) we see that for the purpose of selecting $x_s \sim U_{c-1}$ via $\mathbb{E}_{(x_i,y_i)\sim D}[l(x_i,y_i;M_{L_c})]$, the training loss becomes irrelevant (a constant), so it can be safely ignored. Therefore, we can use the following objective in place of (1):

$$\underset{x_s \sim U_{c-1}}{\arg\min} \ \mathbb{E}_{(x_i \sim U_c, y_i \sim U_c^Y)}[l(x_i,y_i;M_{L_c})], \tag{3}$$

however, akin to that in (1), the label $y_s$ and label set $U_c^Y$ are not available. In order to tackle this problem, below we present Theorem 1 that bridges the gap between a practical but estimated loss and the inaccessible but actual loss.

**Theorem 1.** *Given the true data joint distribution $p_X p_{Y|X}$, where $\mathcal{X}$ is the input space and $\mathcal{Y}$ is the label space, and an estimated data joint distribution $p_X \tilde{p}_{Y|X}$ (assuming we can access the input distribution $p_X$ but not the true label conditional $p_{Y|X}$), any positive and bounded loss function $l(\cdot,\cdot;M) : \mathcal{X} \times \mathcal{Y} \mapsto [0, L]$ for model $M$ must satisfy*

$$\mathbb{E}_{(x,y)\sim p_X p_{Y|X}} l(x,y;M) \leq \mathbb{E}_{(x,\tilde{y})\sim p_X \tilde{p}_{Y|X}} l(x,\tilde{y};M) + 2L \cdot \delta_{TV}(p_X p_{Y|X}, p_X \tilde{p}_{Y|X}), \tag{4}$$

*where $\delta_{TV}(\cdot,\cdot)$ denotes the total variation difference between two distributions.*

The detailed proof for Theorem 1 can be found in Appendix A.1. Conceptually, Theorem 1 shows that the difference between the estimated and true loss in (4), obtained via $\tilde{p}_{Y|X}$ and $p_{Y|X}$, respectively, is bounded by two terms: $i$) the magnitude, $L$, of the loss function, and $ii$) the difference between the real and estimated joint distributions via $\delta_{TV}(p_X p_{Y|X}, p_X \tilde{p}_{Y|X})$. In most cases, for instance when using the standard cross-entropy loss for classification, the total variation is bounded by some power $r$ of the norm of the parameters $\|W\|_2^r$, where $W$ denotes all the parameters of the model, which itself is bounded via the use of a regularization term during the training process (Figueiredo, 2001; Figueiredo & Jain, 2001; Figueiredo, 2003). We provide a detailed proof for the cross-entropy loss in Appendix A.1.

With regard to the second term involving $\delta_{TV}(\cdot, \cdot)$, we propose to generate pseudo labels from $M_{L_{c-1}}$ as a means to intrinsically and effectively sample from the estimated label conditional distribution $\tilde{p}_{Y|X}$ implied by $M_{L_{c-1}}$. Conveniently, for the purpose of sample selection, the second term in (4) becomes approximately a constant by assuming that $\delta_{TV}(p_{X \cup x_s} \tilde{p}_{Y|X \cup x_s}, p_{X \cup x_{s'}} \tilde{p}_{Y|X \cup x_{s'}}) \to 0$, where $X \cup x_s$ and $X \cup x_{s'}$ for $x_s \neq x_{s'}$ are used (in a slight abuse of notation) to denote the estimated conditionals with $L_c = L_{c-1} \cup (x_s, y_s)$ and $L_c = L_{c-1} \cup (x_{s'}, y_{s'})$, respectively, which is reasonable as long as $m_{c-1}$ is not too small. Consequently, combining (4) with (3) results in

$$\underset{x_s \sim U_{c-1}}{\arg\min} \ \mathbb{E}_{(x_i \sim U_c, \tilde{y}_i \sim \tilde{p}_{Y|X=x_i})}[l(x_i, \tilde{y}_i; M_{L_c})]. \tag{5}$$

Unlike (3), we can readily evaluate the loss $l(x_i, y_i; M_{L_c})$ on $U_c$ for every sample in $U_{c-1}$. Below, we will describe how to implement (5) provided that $M_{L_c}$ is not available when selecting $x_s \sim U_{c-1}$.

## 4.2 Estimating Loss without Retraining

Though the AL selection objective introduced in (5) is theoretically justified, its practical application to deep neural network architectures raises two key issues that need to be addressed. The first issue has to do with *efficiency* in that implementing equation (5) necessitates training $M_{L_c}$ for each data candidate $(x_s \sim U_{c-1}, y_s \sim U_{c-1}^Y)$, resulting in the training of $n - m_{c-1}$ distinct models. Unfortunately, the time-consuming nature of training deep neural networks makes this process highly impractical or even prohibitive in some cases. The second problem associated with *labeling* is that obtaining $M_{L_c}$ for each candidate data requires access to the true label $y_s \sim U_{c-1}^Y$, which is unavailable during AL selection cycles. To address these two issues, we propose the following strategies.

To tackle the efficiency problem due to the need of explicitly training $n - m_{c-1}$ models in each AL selection cycle, we leverage *influence functions* (IL) (Hampel, 1974). The influence function serves as a proxy to quantify how the parameters of a model change when a training point, denoted as $(x_s, y_s)$, is weighted by an infinitesimal amount. To provide a fundamental understanding of the influence function, we present a simple example. Let us consider a model, denoted as $M$, trained with a labeled dataset $L = (x_i, y_i)_{i=1}^n$, whose parameters $\hat{\theta} \in \Theta$ are obtained via $\hat{\theta} = \arg\min_{\theta \in \Theta} \sum_{i=1}^n l(x_i, y_i; \theta)$. Now, if we incrementally increase the weight of a specific sample $(x_s, y_s)$ by a small value $\epsilon$, the resulting parameters, denoted as $\hat{\theta}_{\epsilon,(x_s,y_s)}$, can be expressed as: $\hat{\theta}_{\epsilon,(x_s,y_s)} = \arg\min_{\theta \in \Theta} \sum_{i=1}^n l(x_i, y_i; \theta) + \epsilon l(x_s, y_s; \theta)$. The influence function, introduced by Hampel (1974), gives insight into the impact of this incremental weighting of $(x_s, y_s)$, which results in $\hat{\theta}_{\epsilon,(x_s,y_s)}$. Formally, the influence function $IL(x_s, \hat{\theta})$ can be defined as:

$$IL(x_s, \hat{\theta}) = \left. \frac{d\hat{\theta}_{\epsilon,(x_s,y_s)}}{d\epsilon} \right|_{\epsilon=0} = -H_{\hat{\theta}}^{-1} \nabla l(x_s, y_s; \hat{\theta})), \tag{6}$$

where $H_{\hat{\theta}} \stackrel{\text{def}}{=} \frac{1}{n} \sum_{i=1}^n \nabla^2 l(x_i, y_i; \hat{\theta})$ is the Hessian of model $M$. The detailed proof of (6) can be found in Koh & Liang (2017).

In the context of active learning, we aim to obtain model $M_{L_c}$ in (5) by incorporating a sample $(x_s, y_s) \sim U_{c-1}$ into the training dataset $L_{c-1}$. Assuming that we already have the parameters of the model trained with $L_{c-1}$, denoted as $\theta_{M_{L_{c-1}}}$, which were obtained via $\theta_{M_{L_{c-1}}} = \arg\min_{\theta \in \Theta} \frac{1}{m_c} \sum_{i=1}^{m_c} l(x_i, y_i; \theta)$. The parameters

of $M_{L_c}$ can be approximated as

$$\theta_{M_{L_c}} \approx \mathrm{argmin}_{\theta \in \Theta} \ \frac{1}{m_c} \sum_{i=1}^{m_c} l(x_i, y_i; \theta) + \frac{1}{m_c} l(x_s, y_s; \theta).$$

This approximation suggests that incorporating a sample $(x_s, y_s) \sim U_{c-1}$ into the training dataset $L_{c-1}$ can be thought as to incrementally weighting the sample $(x_s, y_s)$ by $m_c^{-1}$. Leveraging the influence function as in (6), we can estimate the parameters of $M_{L_c}$, denoted as $\theta_{M_{L_c}}$, efficiently, that is via $\theta_{M_{L_c}} \approx \theta_{M_{L_{c-1}}} + IL(x_s, \theta_{M_{L_{c-1}}})$. Conveniently, this allows us to efficiently obtain the parameters for $M_{L_c}$ in (5) without the need to explicitly train $M_{L_c}$ with $L_{c-1} \cup (x_s, y_s)$ for each data candidate $(x_s \sim U_{c-1}, y_s \sim U_{c-1}^Y)$. Additional details about the theory and efficient implementation of the IL can be found in Koh & Liang (2017).

To address the labeling problem associated with lacking access to $y_s \sim U_{c-1}^Y$, we consider three approaches. The first one, which we refer to as *single-IL*, uses pseudo labels generated by $M_{L_{c-1}}$, *i.e.*, by simply replacing $y_s$ with the (most likely) pseudo label $\hat{y}_s = \arg\max \ M_{L_{c-1}}(x_s)$. This can be formulated as

$$\mathbb{E}_{(x_i \sim U_c, \tilde{y}_i \sim \tilde{p}_{Y|X})}[l(x_i, \tilde{y}_i; M_{L_c})] \approx \mathbb{E}_{(x_i \sim U_c, \tilde{y}_i \sim \tilde{p}_{Y|X})}[l(x_i, \tilde{y}_i; M_{L_{c-1}} \cup (x_s, \hat{y}_s))]. \tag{7}$$

Though conceptually simple, this method may introduce severe approximation bias if $y_s \neq \hat{y}_s$. Alternatively, to obtain a relaxed approximation of $\mathbb{E}_{(x_i \sim U_c, \tilde{y}_i \sim \tilde{p}_{Y|X})}[l(x_i, y_i; M_{L_c})]$ for each $x_s$, we account for all class labels for each $x_s$, rather than relying solely on the single most probable (maximum likelihood) class $\hat{y}_s$ in (7). One straightforward approach is to weight all possible classes for $x_s$ using the probabilistic predictions from $M_{L_{c-1}}(x_s)$. We refer to this approach as *expectation-IL* and formulate it as

$$\mathbb{E}_{(x_i \sim U_c, \tilde{y}_i \sim \tilde{p}_{Y|X})}[l(x_i, \tilde{y}_i; M_{L_c})] \approx \sum_{k=1}^{K} p_k(x_s) \cdot \mathbb{E}_{(x_i \sim U_c, \tilde{y}_i \sim \tilde{p}_{Y|X})}[l(x_i, \tilde{y}_i; M_{L_{c-1} \cup (x_s, \tilde{y}_s^k)})], \tag{8}$$

where $\tilde{y}_s^k$ denotes setting the label for $x_s$ as class $k$, $p_k(x_s)$ is the predicted probability (weight) that $x_s$ is of class $k$, which is obtained from $M_{L_{c-1}}$, and $K$ is the number of classes. Note that this is conceptually equivalent to considering all possible outcomes for $x_s$, but weighting them according to how likely they are under the current version of the model, *i.e.*, $M_{L_{c-1}}$.

In practice, we found that (8) has the potentially significant problem that for unlikely classes, $p_k(x_s)$ is likely to be nonzero, thus introducing estimation noise. Moreover, we are mostly interested in classes for which $p_k(x_s)$ is large or at least comparable to $\max \ M_{L_{c-1}}(x_s)$. With this in mind, we propose an alternative solution we call *truncated-IL*, which involves considering several high-probability class labels for each $x_s$ instead of all classes. More formally we write

$$\mathbb{E}_{(x_i \sim U_c, \tilde{y}_i \sim \tilde{p}_{Y|X})}[l(x_i, \tilde{y}_i; M_{L_c})] \approx \sum_{k=1}^{K} \pi(p_k(x_s)) \cdot p_k(x_s) \cdot \mathbb{E}_{(x_i \sim U_c, \tilde{y}_i \sim \tilde{p}_{Y|X})}[l(x_i, \tilde{y}_i; M_{L_{c-1} \cup (x_s, \tilde{y}_s^k)})], \tag{9}$$

from which we see that compared with (8), (9) has a indicator function $\pi(\cdot)$ defined as

$$\pi(p_k(x_s)) = \begin{cases} 1 & \text{if } p_k(x_s) > \tau \\ 0 & \text{otherwise} \end{cases}, \tag{10}$$

where $\tau$ is a threshold value set to filter out highly unlikely class assignments for $x_s$. For all experiments, we set $\tau = \frac{4}{K}$, which we found empirically gives the best overall performance. Nevertheless and for illustration, we provide a comparison of the results obtained by using different values of $\tau$ in one specific active learning scenario in Appendix A.3.

Having addressed the challenges involved in implementing (5) that are associated with efficiency and labeling, we have developed a single-sample sequential DL-based AL algorithm. Conceptually, we will rank all unlabeled samples based on their ability to minimize the model loss on the current unlabeled dataset using *single-IL* in (7), *expectation-IL* in (8) or *truncated-IL* in (9), and then select the top-ranked sample at each AL selection cycle, however, as discussed below, this approach may not be practical for DL models.

---

**Algorithm 1** RALIF: Reliable active learning via influence functions with *truncated-IL* selection.

---

**Input:** $L_0$: Initial labeled dataset, $U_0$: initial unlabeled dataset, $M$: model, $C$: AL cycles, $B$: query set, $P$: candidate set, $\phi_{M_{L_{c-1}}}(\cdot)$: convolutional encoder, and $\tau$: truncation hyperparameter.

1: **for** $c = 1, \ldots, C$ **do**
2:     Train model $M_{L_{c-1}}$ with labels set $L_{c-1}$
3:     $|P| = 0.1|U_{c-1}|$
4:     $\forall x_i \sim U_c$ generate pseudo-labels $\tilde{y}_i = \arg\max M_{L_{c-1}}(x_i)$
5:     **for** $x_s \in U_{c-1}$ **do**
6:         Generate probabilistic predictions $\{p_k(x_s)\}_{k=1}^K$ from $M_{L_{c-1}}(x_s)$
7:         Calculate $\mathbb{E}_{(x_i \sim U_c, \tilde{y}_i \sim \tilde{p}_{Y|X})}[l(x_i, \tilde{y}_i; M_{L_c})]$ via (9)
8:     **end for**
9:     Select top-$|P|$ samples $\{x_i\}_{|P|}$ with minimum $\mathbb{E}_{(x_i \sim U_c, \tilde{y}_i \sim \tilde{p}_{Y|X})}[l(x_i, \tilde{y}_i; M_{L_c})]$
10:     Run *k-MEANS++* on $\{\phi_{M_{L_{c-1}}}(x_i)\}_{|P|}$ to select $|B|$ diverse samples
11:     $L_c = L_{c-1} \cup \{(x_s, y_s)\}_{s \in B}$
12: **end for**
**Return:** Final labeled dataset $L_C$ (of size $L_0 + C|B|$)

---

### 4.3 Batch Selection AL for DL

In the previous subsection we introduced a practical single-sample sequential deep learning active learning algorithm. However, when dealing with deep neural networks, adding only one sample at the time to the training dataset is not likely to meaningfully affect performance between AL cycles[2], but will definitely increase computational requirements to a point that in some scenarios it may render AL prohibitive. Therefore, we opt for batch selection in each AL cycle, which is a common practice in active learning selection mechanisms for deep neural networks (Kirsch et al., 2019; Sener & Savarese, 2017; Wang et al., 2022).

From (5) we can conceptualize that given a sample $x_s$ that minimizes the loss, there are likely other samples in $U_{c-1}$ that being very similar to $x_s$ also have comparably small losses. However, by virtue of being similar they are also likely to be redundant in their ability to improve model $M$. Though such redundancy is not likely to negatively impact performance, it certainly affects efficiency, by requiring unnecessary labeling efforts. This phenomenon of batch AL approaches has been described before (Beluch et al., 2018; Ash et al., 2019; Wang et al., 2022), and several approaches have been proposed to mitigate this problem. For example, Beluch et al. (2018) suggested selecting samples from a subset of the unlabeled dataset rather than considering the entire dataset to avoid including highly similar samples in a selected batch.

To alleviate such issue, we propose a two-step approach. First, we utilize *single-IL* in (7), *expectation-IL* in (8) or *truncated-IL* in (9) to select a pool of candidate samples with the smallest expected losses on the unlabeled dataset. In practice, we set the size of the candidate pool $P$ to be 10% of the size of the unlabeled set, *i.e.*, $P = 0.1 \times |U_c|$. Second, to ensure the diversity of the final selected samples, we cluster the $P$ candidates into $B$ clusters, where $B$ refers to the budget size for each AL selection cycle, and then select from $P$ the centroids of these clusters as the samples selected for labeling. In order to ensure that selected samples are representative of the entire pool of candidates, we use the features extracted from the convolutional encoder as the representation space for clustering. Specifically, we use the *k-MEANS++* (Arthur & Vassilvitskii, 2007) algorithm, a variant of the popular $K$-means (MacQueen, 1967) algorithm. This particular clustering algorithm was chosen because work by Arthur & Vassilvitskii (2007); Ash et al. (2019) demonstrated that *k-MEANS++* has a fast convergence speed while generating relatively diverse clustering centroids, which encourages the diversity of the set of selected samples.

In practice, one cycle of the RALIF framework consists on selecting a diverse set of samples with the smallest loss on the unlabeled data given in (7), (8) or (9), while using pseudo labels from the current trained model as a proxy for the true, but the unavailable, labels. The complete process for *truncated-IL* in (9) is summarized in Algorithm 1. Further, we provide a more complete algorithm in Appendix A.2, which includes all selection approaches, *i.e.*, *single-IL* in (7), *expectation-IL* in (8), and *truncated-IL* in (9).

---

[2]In fact, it may be detrimental in some cases due to the use of stochastic gradient learning approaches.

## 5 Experiments

Our experiments aim to demonstrate the reliability of RALIF on various selection scenarios. Below, we first introduce the baseline algorithms with which we compare. Then, we describe the different AL selection scenarios that we consider in our experiments, along with the model training details. Finally, we will present the results of our experiments, highlighting the performance of the proposed RALIF framework.

**Baselines** We consider the following baselines in our experiments: *i*) Random: selecting a random batch of unlabeled samples at each selection cycle, which serves as the naive, *no skill*, baseline for active learning; *ii*) Uncertainty: we leverage the most commonly used uncertainty-based selection method introduced in Beluch et al. (2018), which involves selecting unlabeled samples with the highest entropy based on the classification probabilities produced by the model trained with the labeled dataset; *iii*) CoreSet (Sener & Savarese, 2017): utilizing the model trained with the labeled dataset to extract features for all unlabeled samples, then applying the *K-Center-Greedy* (Wolf, 2011) algorithm on such feature space to select a diverse batch of unlabeled samples; *iv*) BADGE (Ash et al., 2019): selecting unlabeled samples by applying the *K-MEANS++* algorithm on the gradient norms calculated for all unlabeled samples based on the pseudo labels predicted by the model trained with the labeled data; and *v*) GradNorm (Wang et al., 2022): selecting unlabeled samples with the largest gradient norms by using their entropy as loss for backpropagation. Note that this is the same entropy used in Uncertainty.

The five baselines we consider in our experiments cover the most common types of active learning algorithms, namely, Random selection (as a naive baseline for AL), uncertainty-based selection (Uncertainty), diversity-based selection (Coreset), hybrid strategies (BADGE, which combines uncertainty and diversity), and DL-specific selection (GradNorm). In comparison to these baselines, RALIF can be considered a hybrid DL-specific AL algorithm as it combines the DL-specific selection (utilizing influence functions), but encouraging the selection of a set of diverse samples that are likely to minimize the loss incurred by the model on the unlabeled dataset. By comparing RALIF with these baselines, we will also demonstrate how it fares with the state-of-the-art methods, *i.e.*, Ash et al. (2019); Wang et al. (2022).

**AL Scenarios** To simulate various selection scenarios, we consider three main determinants in the active learning selection process, namely, dataset, model architecture, and budget size at each selection cycle. In our experiments, we use three datasets: Cifar10 (Krizhevsky et al., 2009), Cifar100 (Krizhevsky et al., 2009), and iNaturalist (14 super classes) (Van Horn et al., 2018), representing simple, medium complexity, and imbalanced datasets, respectively. In terms of model architectures, we consider three options: ResNet18 from scratch denoted as ResNet18 (scratch), ResNet18 with pretrained weights from ImageNet (Deng et al., 2009) denoted, and a simple linear classifier (a fully connected layer) with a fixed but powerful encoder (Clip-Vit-patch16) denoted as CLIP. We use the standard ResNet18 (He et al., 2016) architecture for the pretrained ResNet18, the standard Clip-Vit-patch16 (Radford et al., 2021) for CLIP, and a customized version of ResNet18 for ResNet18 (scratch)[3], where the kernel size of the first convolution layer is modified from 7 to 3 and the max pooling layer is removed. Moreover, we also consider two different budget sizes: 1000 or 2000 samples at each selection cycle. In Appendix A.5, we included additional active learning scenarios considering other datasets and models for a more comprehensive evaluation.

**Training Details** We set different initial labeled dataset sizes for each selection scenario based on model capacity and the complexity of the dataset. For the CLIP model, which has shown excellent performance with a smaller training dataset size compared to other models, we set the initial labeled dataset size to 100 samples. This choice is made to ensure that the initial accuracy is not excessively high, allowing us to observe the effectiveness of the AL algorithms. For all other models, we set the initial labeled dataset size to 1000 samples for Cifar10 and 6000 samples for Cifar100 and iNaturalist. This decision takes into account the higher complexity of the iNaturalist and Cifar100 datasets relative to Cifar10. We trained our models using stochastic gradient descent (Sutskever et al., 2013) with a weight decay of $10^{-4}$ and a momentum of 0.9. The initial learning rate, learning rate scheduler, and training epochs were set based on the model capacity. For pretrained ResNet18 and CLIP, we employed CosineAnnealingLR (Loshchilov & Hutter, 2016)

---

[3]Source code available at https://github.com/kuangliu/pytorch-cifar.

Table 1: Ablation study results. We present the mean and standard deviation of the performance metrics for all algorithms (different selection strategies and diversity selection) over three trials. Best results are highlighted in green.

| Candidate Selection | Diversity $k$-MEANS++ | Cycle First ($1^{st}$) | Last ($9^{th}$) | AUCA Ratio |
|---|---|---|---|---|
| *Random* | | $85.31 \pm 0.29$ | $92.29 \pm 0.09$ | $1.000 \pm 0.000$ |
| *single-IL* | | $85.60 \pm 0.60$ | $93.86 \pm 0.04$ | $1.014 \pm 0.000$ |
| *expectation-IL* | | $85.62 \pm 0.38$ | $93.00 \pm 0.09$ | $1.004 \pm 0.001$ |
| *truncated-IL* | | $86.15 \pm 0.20$ | $94.10 \pm 0.09$ | $1.017 \pm 0.002$ |
| *single-IL* | ✓ | $86.23 \pm 0.49$ | $94.05 \pm 0.04$ | $1.017 \pm 0.001$ |
| *expectation-IL* | ✓ | $85.66 \pm 0.18$ | $92.92 \pm 0.16$ | $1.006 \pm 0.001$ |
| *truncated-IL* | ✓ | $86.92 \pm 0.15$ | $94.35 \pm 0.05$ | $1.020 \pm 0.002$ |

as the learning rate scheduler and set the initial learning rate to 0.01. The pretrained ResNet18 and CLIP models were trained for 200 and 100 epochs, respectively, the latter accounting for the excellent performance of the CLIP model. For ResNet18 (scratch), following the suggestions in He et al. (2016), we set the initial learning rate to 0.1 and decrease it by a factor of 10 at the 160-th epoch, for a total of 200 training epochs. In our experiments, we consider basic data augmentations, including random horizontal flip, random crop, and normalization. For the input images, if the model is ResNet18 (scratch), we resize the inputs to $32 \times 32$, otherwise, we resize the inputs to $224 \times 224$. This choice is based on the different architectures of the models and their specific requirements for input sizes. For each selection scenario, we run each algorithm three times over 9 AL selection cycles and report the mean and standard deviation of the results as the final performance of the algorithms. Source code is available at `https://github.com/mx41-m/Active-Learning.git`

**Evaluation Metrics**  In our experiments, we use classification accuracy, denoted as ACC, as the primary evaluation metric for comparing the performance of the AL algorithms. Specifically, we train a model with the labeled dataset selected by each AL algorithm at each selection cycle and evaluate the classification accuracy of the model on the corresponding test dataset. To provide a more comprehensive metric of the performance of each AL algorithm over the entire selection process (9 selection cycles), we introduce a summary performance metric we call AUCA Ratio representing the ratio between the area under the classification accuracy curve over all selection cycles for a model and the random selection approach. The AUCA Ratio is calculated using the trapezoidal rule (Yeh et al., 2002). Note that an AUCA Ratio larger than 1 indicates that the corresponding AL algorithm achieves a consistent improvement over the random selection baseline throughout the entire selection process.

## 5.1 Ablation Study

The proposed RALIF consists of mainly two components. We first utilize results from Theorem 1 and the influence function formulation in (6) via (5) to select the most likely informative and useful sample candidates for labeling. We then use the *k-MEANS++* algorithm on the features of the selected pool of candidates to down-select a batch encouraging sample diversity. We consider three options for the first component of RALIF: *single-IL* in (7), *truncated-IL* in (9), and *truncated-IL* in (8). For the second component, we can choose (or not) to favor diversity by applying the *k-MEANS++*. If we choose not to apply *k-MEANS++*, we simply select a batch of the top-ranked $B$ samples obtained using the ranked loss from the first component.

To investigate the contribution of different choices, we conduct an ablation study on a general selection scenario: using the Cifar10 dataset, the pretrained ResNet18 model architecture, and a budget size of 1000. We compare performance metrics, ACC and AUCA Ratio, after the first and last selection cycles, first and ninth, respectively. Table 1 underscores that each component of our algorithm contributes to the selection process by prioritizing more useful samples compared to the random selection baseline. From Table 1, we also observe that, compared with *single-IL* and *truncated-IL*, *expectation-IL* is less successful at selecting useful samples. This is likely to be caused by samples with small but nonzero probabilities assigned during predictions that negatively bias their influence estimation, thus undermining the selection

process. Complementary, the results suggest that applying the *k-MEANS++* algorithm in general improves the performance of the the AL algorithm by alleviating information redundancy during the selection process. Overall, the combination of *truncated-IL* and *k-MEANS++* achieved the best performance, so we will use this combination, simply referred to as *truncated-IL* for further comparisons with all AL baselines. Additionally, we have included the results of using different diversity operations in Appendix A.4, providing further insights into the impact of these on the performance of our approach.

## 5.2 Performance Comparison with Baselines

We conduct a comprehensive comparison of the proposed *truncated-IL* algorithm (with diversity selection) with other baselines in six simulated benchmark scenarios with different datasets (Cifar10, Cifar100 and iNaturalist), batch size (100, 1000 and 2000) and DL model (pretrained RestNet18, ResNet18 from scratch and pretrained CLIP). Table 2 provides a summary of the results, focusing on ACC and AUCA Ratio comparisons for the first and last AL selection cycles. For more detailed ACC comparisons across all selection cycles and boxplots summarizing the AUCA Ratio for different trials see Appendix A.8. In summary, the results presented in Table 2 clearly demonstrate that RALIF with *truncated-IL* selection consistently achieved the best performance across all scenarios, highlighting its reliability and effectiveness.

More specifically, among all the baselines considered, BADGE fared as the second-best approach in three scenarios, followed by Grad-Norm in two scenarios, and Uncertainty in one scenario. However, it is important to note that all baselines performed worse than the Random baseline in at least one selection cycle (highlighted in red in Table 2). In contrast, RALIF with *truncated-IL* selection always outperformed Random. For instance, when considering the Cifar10 dataset with a budget size of 1000 and the ResNet18 (scratch) model, all baselines performed worse than Random in the first selection cycle. Notably, the Coreset algorithm exhibited worse performance than the Random baseline in two scenarios, while Uncertainty performed worse than Random in one scenario. As a summary of the experiments across different settings, we also present average metrics, from which we can see the consistency of RALIF relative to the competing approaches. These results highlight the reliability of RALIF compared to baselines including state-of-the art methods. Further, to stress out the significance of the performance differences between RALIF and other approaches, we have included detailed results of statistical significance tests in Appendix A.7. Moreover, these significance results have been summarized in the bottom row of Table 2, which indicates the proportion of times that the proposed RALIF achieves statistically significant performance differences relative to other approaches across all AL scenarios, including the additional AL scenarios presented in Appendix A.5.

## 5.3 Qualitative results

To gain further insights into the samples selected by different AL approaches, we present visualizations of selected samples using UMAP embeddings for the iNaturalist dataset with AL approaches with a budget of 100 and a fixed CLIP encoder. Figure 2 shows UMAP visualizations of unlabeled and selected samples (in red) for Uncertainty, BADGE, and RALIF. We chose Uncertainty and BADGE for visualization because the former is the most commonly used AL algorithm, and the latter is the second-best AL algorithm among the baselines based on the experimental results in Table 2. In the figure, each point represents a sample, different colors indicate true classes (first two rows) or entropy (third row). Note that true labels are not available to the AL methods.

From Figure 2, several observations can be made. First, compared to the other two AL algorithms, the samples selected by Uncertainty appear to cluster in a smaller area consistent with high uncertainty (entropy), and with most of them closely packed together. This suggests that the Uncertainty algorithm tends to select similar samples in each iteration, leading to redundancy in the selected data and potentially undermining its performance, thus underscoring the benefits of diversity sampling. However, results for Coreset in Table 2 indicate that solely relying diversity during active learning selection cycles can lead to worse performance. Alternatively, BADGE combines uncertainty and diversity in its strategy. Examining Figure 2, we observe that the samples selected by BADGE are more diverse compared to those selected by Uncertainty. This indicates that BADGE effectively avoids selecting samples that are clustered together and rather focuses on selecting more diverse samples. From Figure 2, we also see that RALIF, similar to BADGE, successfully

Table 2: Comparison with baselines in different scenarios. *B*: budget size at each selection cycle. Note that for RALIF we use *truncated-IL* for candidate selection and *K-MEANS++* for diversity down-selection. We present the mean and standard deviation of the results for all algorithms over three trials. The best and worse than *Random* results are highlighted in green and red, respectively. Underlined values indicate instances where the performance difference between RALIF and other approaches is *not* statistically significant, *i.e.*, the *p*-value for the significance test exceeds the 0.05 threshold. The Average row summary the overall performance averaged of different algorithms across all active learning (AL) scenarios, including the additional AL scenarios presented in Appendix A.5.

| | Setting | | Methods | Cycles | | AUCA Ratio |
|---|---|---|---|---|---|---|
| Datasets | $B$ | models | | First($1^{st}$) | Last ($9^{th}$) | |
| Cifar10 | 1000 | *ResNet18 (scratch)* | Random | $61.33 \pm 1.20$ | $87.04 \pm 0.11$ | $1.000 \pm 0.000$ |
| | | | Uncertainty | $60.27 \pm 3.06$ | $90.39 \pm 0.08$ | $1.025 \pm 0.008$ |
| | | | Coreset | $54.07 \pm 2.68$ | $84.80 \pm 0.68$ | $0.948 \pm 0.006$ |
| | | | BADGE | $60.21 \pm 0.88$ | $89.79 \pm 0.12$ | $1.024 \pm 0.006$ |
| | | | Grad-Norm | $57.90 \pm 3.78$ | $90.08 \pm 0.25$ | $1.027 \pm 0.002$ |
| | | | RALIF | $62.05 \pm 0.75$ | $90.53 \pm 0.13$ | $1.041 \pm 0.003$ |
| Cifar10 | 1000 | *ResNet18* | Random | $85.31 \pm 0.29$ | $92.29 \pm 0.09$ | $1.000 \pm 0.000$ |
| | | | Uncertainty | $85.55 \pm 0.28$ | $94.29 \pm 0.24$ | $1.016 \pm 0.003$ |
| | | | Coreset | $85.59 \pm 0.11$ | $93.04 \pm 0.12$ | $1.008 \pm 0.002$ |
| | | | BADGE | $86.66 \pm 0.16$ | $94.32 \pm 0.06$ | $1.019 \pm 0.001$ |
| | | | Grad-Norm | $86.60 \pm 0.07$ | $94.19 \pm 0.07$ | $1.019 \pm 0.003$ |
| | | | RALIF | $86.92 \pm 0.15$ | $94.35 \pm 0.05$ | $1.021 \pm 0.002$ |
| Cifar10 | 2000 | *ResNet18* | Random | $87.56 \pm 0.25$ | $94.06 \pm 0.04$ | $1.000 \pm 0.000$ |
| | | | Uncertainty | $88.64 \pm 0.29$ | $95.57 \pm 0.03$ | $1.019 \pm 0.001$ |
| | | | Coreset | $87.58 \pm 0.17$ | $94.68 \pm 0.20$ | $1.007 \pm 0.001$ |
| | | | BADGE | $88.88 \pm 0.19$ | $95.69 \pm 0.16$ | $1.018 \pm 0.001$ |
| | | | Grad-Norm | $88.46 \pm 0.11$ | $95.70 \pm 0.03$ | $1.018 \pm 0.001$ |
| | | | RALIF | $88.96 \pm 0.05$ | $95.72 \pm 0.07$ | $1.020 \pm 0.001$ |
| Cifar100 | 1000 | *ResNet18* | Random | $64.50 \pm 0.55$ | $71.64 \pm 0.13$ | $1.000 \pm 0.000$ |
| | | | Uncertainty | $64.81 \pm 0.13$ | $72.41 \pm 0.25$ | $1.011 \pm 0.003$ |
| | | | Coreset | $64.86 \pm 0.10$ | $71.67 \pm 0.37$ | $1.003 \pm 0.001$ |
| | | | BADGE | $65.26 \pm 0.16$ | $73.21 \pm 0.19$ | $1.013 \pm 0.001$ |
| | | | Grad-Norm | $65.27 \pm 0.34$ | $73.11 \pm 0.31$ | $1.010 \pm 0.003$ |
| | | | RALIF | $65.17 \pm 0.08$ | $73.26 \pm 0.08$ | $1.015 \pm 0.002$ |
| iNaturalist | 1000 | *ResNet18* | Random | $85.17 \pm 0.07$ | $87.10 \pm 0.07$ | $1.000 \pm 0.000$ |
| | | | Uncertainty | $85.14 \pm 0.08$ | $87.45 \pm 0.08$ | $1.002 \pm 0.000$ |
| | | | Coreset | $85.43 \pm 0.03$ | $87.45 \pm 0.07$ | $1.003 \pm 0.000$ |
| | | | BADGE | $85.53 \pm 0.14$ | $88.40 \pm 0.01$ | $1.010 \pm 0.001$ |
| | | | Grad-Norm | $85.56 \pm 0.05$ | $88.47 \pm 0.08$ | $1.010 \pm 0.001$ |
| | | | RALIF | $85.66 \pm 0.05$ | $88.63 \pm 0.12$ | $1.012 \pm 0.001$ |
| iNaturalist | 100 | *CLIP* | Random | $90.85 \pm 0.90$ | $93.98 \pm 0.18$ | $1.000 \pm 0.000$ |
| | | | Uncertainty | $89.13 \pm 1.00$ | $93.72 \pm 0.20$ | $0.994 \pm 0.007$ |
| | | | Coreset | $89.64 \pm 0.90$ | $93.44 \pm 0.18$ | $0.990 \pm 0.004$ |
| | | | BADGE | $92.66 \pm 0.32$ | $95.17 \pm 0.08$ | $1.015 \pm 0.004$ |
| | | | Grad-Norm | $92.71 \pm 0.38$ | $95.05 \pm 0.02$ | $1.015 \pm 0.004$ |
| | | | RALIF | $92.82 \pm 0.26$ | $95.17 \pm 0.01$ | $1.015 \pm 0.002$ |
| Average | | | Random | $81.67 \pm 10.29$ | $89.58 \pm 6.92$ | $1.000 \pm 0.000$ |
| | | | Uncertainty | $81.71 \pm 10.58$ | $91.02 \pm 7.11$ | $1.012 \pm 0.010$ |
| | | | Coreset | $80.76 \pm 11.93$ | $89.67 \pm 7.29$ | $0.996 \pm 0.020$ |
| | | | BADGE | $82.62 \pm 10.94$ | $91.30 \pm 6.90$ | $1.016 \pm 0.005$ |
| | | | Grad-Norm | $82.26 \pm 11.50$ | $91.30 \pm 6.92$ | $1.016 \pm 0.005$ |
| | | | RALIF | $83.05 \pm 10.66$ | $91.53 \pm 6.93$ | $1.020 \pm 0.008$ |
| RALIF is statistically significant | | | | $33/45(73.33\%)$ | $34/45(75.56\%)$ | $19/45(42.22\%)$ |

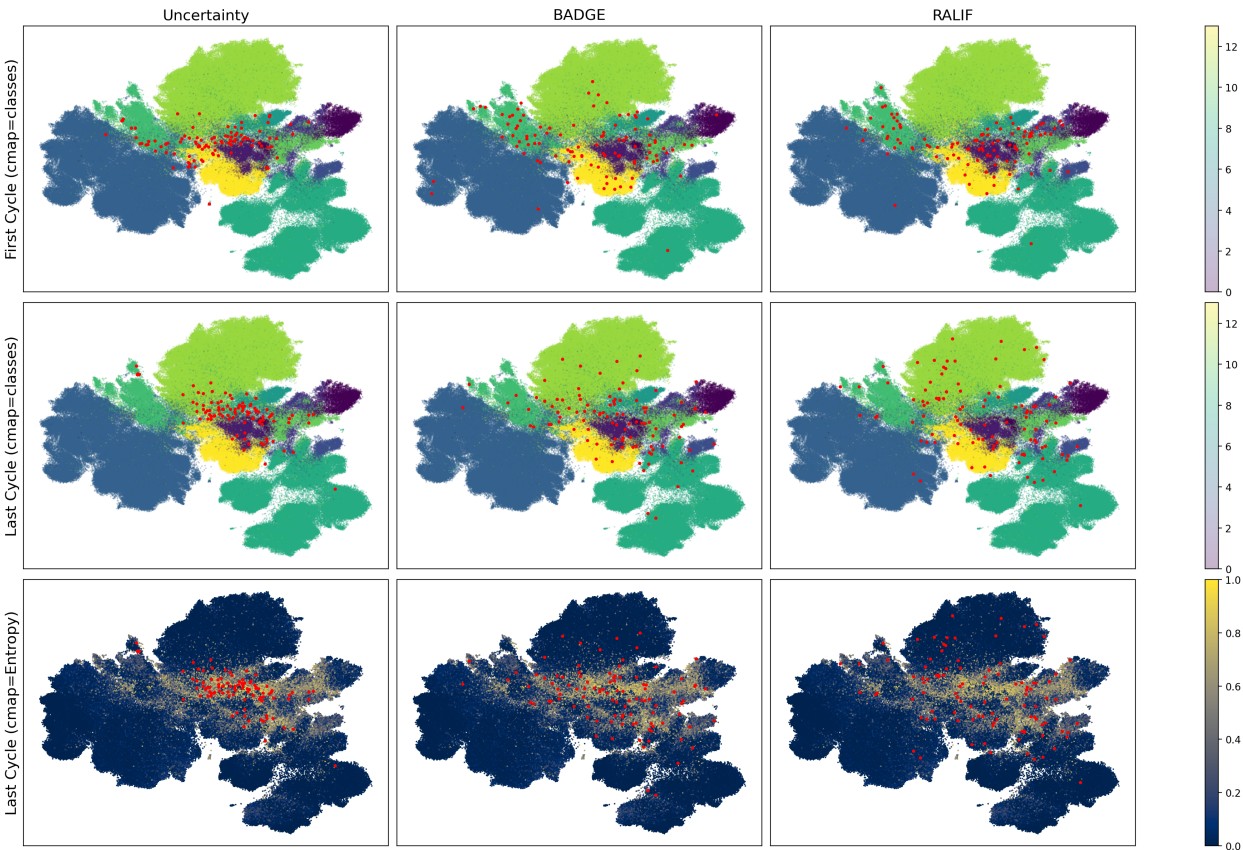

Figure 2: UMAP embeddings for the first and last selection cycles of different active learning approaches with a fixed CLIP encoder on the iNaturalist data. The first and second rows correspond to the first and last selection cycle, respectively, and with colors indicating true classes for each data point. The bottom row corresponds to the last cycle with colors denoting the normalized entropy for each data point. From left to right, the visualizations are for Uncertainty, BADGE, and RALIF. Red dots highlight samples selected for labeling by each approach.

avoids selecting samples that cluster together and selects more diverse samples than Uncertainty. Notably, RALIF seems to select samples more sparsely distributed relative to BADGE, underscoring the effectiveness of the *truncated-IL* selection approach, by minding that both BADGE and RALIF use the same diversity strategy. Further, comparing sample selections for the first and last cycle, we see that Uncertainty naturally favors tightly packed samples with high entropy, BADGE consistently selects sparsely distributed samples, whereas RALIF initially favors more tightly packed samples and then (as the model improves) starts favoring more diverse samples, thus differentiating itself from both Uncertainty and BADGE.

## 6  Conclusion

In this paper, we proposed a new AL framework for DL architectures called RALIF, which is motivated by theoretical observations on the complete-data loss of a classification model. We established that samples minimizing the model loss on the unlabeled dataset with generated pseudo labels bound from above the loss on the entire dataset with true labels. To address the challenges associated with evaluating the impact of candidate samples on the loss of the model on the unlabeled dataset, we introduce three efficient selection strategies, namely, *single-IL*, *expectation-IL* and *truncated-IL*. These strategies allow the quantification of the contribution of candidate samples to the loss on the entire dataset without the need for explicit model

retraining. Further, we incorporated a diversity prioritization based on *K-MEANS++* similar to BADGE (Ash et al., 2019), to improve the diversity of selected samples when performing batch selection.

We evaluated the reliability and effectiveness of RALIF through comprehensive experiments across various scenarios. The experimental results clearly demonstrated that RALIF consistently outperformed the random baseline and other DL-based AL algorithms in terms of classification performance. Moreover, we provided qualitative results that highlighted RALIF's ability to trade-off high-uncertainty and diversity during the active selection process.

One limitation of RALIF is that the quality of the pseudo labels generated from the model $M$ can affect its performance. There are two main reasons for this behavior. First, RALIF utilizes the model and its predictions as an approximation to the conditional distribution $\tilde{p}_{Y|X}$ in Theorem A.1. If the quality of the model and thus its pseudo labels is poor, the total variation difference $\delta_{TV}(\cdot, \cdot)$ between the estimated label conditional distribution $\tilde{p}_{Y|X}$ and the true label conditional distribution $p_{Y|X}$ will be large. This can result in a loose bound in Theorem A.1, rendering it less effective for AL. Second, RALIF uses pseudo labels to approximate the influence function in (7). If the quality of the pseudo labels is low, the approximated influence loss for each candidate will be inaccurate. This can affect the selection process and potentially lead to suboptimal results. However, it is important to note that this limitation is not exclusive to RALIF but also applies to other AL baselines since they also dependent on the quality and performance of the model at each selection cycle. For example, when considering Cifar10 with a budget size of 1000 and the ResNet18 model trained from scratch, the results in Table 2 demonstrate that all other AL baselines perform worse than the Random baseline in the first cycle, when the initial model has poor performance. Another limitation of RALIF is that it is currently designed specifically for classification tasks, thus it may not be directly applicable to other types of tasks, such as object detection or natural language processing problems. Consequently, further research and adaptation would be needed to extend RALIF to such tasks.

In future work, we plan to extend RALIF to real-world datasets that exhibit specific label structures, such as datasets with hierarchically organized labels. These datasets pose unique challenges for AL algorithms, as the label hierarchy introduces additional dependencies and relationships among the samples. Moreover, it will be interesting to explore how RALIF can be used to address related tasks such as few-shot learning and test-time domain generalization.

## Acknowledgements

The authors would like to thank the editor and the anonymous reviewers for their insightful comments. This research was supported by NIH/NINDS 1R61NS120246, NIH/NIDDK R01- DK123062, and ONR N00014-18-1-2871-P00002-3.

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

## A  Appendix

### A.1  Proof and explanation of Theorem 1

*Proof.* Let $p_{X,Y} = p_X p_{Y|X}, \tilde{p}_{X,Y} = p_X \tilde{p}_{Y|X}$,

$$
\begin{aligned}
\left| \mathbb{E}_{(x,y)\sim p_X p_{Y|X}} l(x,y) - \mathbb{E}_{(x,\tilde{y})\sim p_X \tilde{p}_{Y|X}} l(x,\tilde{y}) \right| &= \left| \int_{X\times Y} \left(p_{X,Y}(x,y) - \tilde{p}_{X,Y}(x,y)\right) l(x,y) dx dy \right| \\
&\leq \int_{X\times Y} |p_{X,Y}(x,y) - \tilde{p}_{X,Y}(x,y)| \cdot |l(x,y)| \, dx dy \\
&\leq L \cdot \int_{X\times Y} |p_{X,Y}(x,y) - \tilde{p}_{X,Y}(x,y)| \, dx dy \\
&= 2L \cdot \delta_{TV}(p_X p_{Y|X}, p_X \tilde{p}_{Y|X}).
\end{aligned}
\tag{11}
$$

$\square$

In practice, the upper bound $L$ of the loss functions can usually be established given that the network inputs are usually bounded and the network weights are regularized during training. An example is the cross-entropy loss function used for classification. Denote the (before softmax activation) network output as $\mathbf{x} \in \mathbb{R}^K$ with elements $x_k$, *i.e.*, $\mathbf{x} = (x_k)_{1\leq k\leq K}$ and the ground truth label vector as $\mathbf{y} \in [0,1]^K$ with elements $y_k$, *i.e.* $\mathbf{y} = (y_k)_{1\leq k\leq K}$, and $\sum_{k=1}^K y_k = 1$. The cross-entropy loss is defined as:

$$
\mathcal{L}_{CE}(\mathbf{x}, \mathbf{y}) = -\sum_{k=1}^K y_k \log p_k,
\tag{12}
$$

where $p_k$ is the post-softmax prediction score for class $k$, *i.e.*,

$$
p_k = \frac{\exp(x_k)}{\sum_{j=1}^K \exp(x_j)}.
\tag{13}
$$

We can show that

$$
\begin{aligned}
\frac{\mathcal{L}_{CE}(\mathbf{x}, \mathbf{y})}{\|\mathbf{x}\|_2} &= \frac{1}{\|\mathbf{x}\|_2} -\sum_{k=1}^K y_k \log \frac{\exp(x_k)}{\sum_{j=1}^K \exp(x_j)} = \frac{1}{\|\mathbf{x}\|_2} \sum_{k=1}^K y_k \log \frac{\sum_{j=1}^K \exp(x_j)}{\exp(x_k)} \\
&\leq \frac{1}{\|\mathbf{x}\|_2} \sum_{i=1}^K y_k \log \frac{K \cdot \exp\left(\max_{1\leq j\leq K} x_j\right)}{\exp(x_k)} \leq \frac{1}{\|\mathbf{x}\|_2} \sum_{i=1}^K y_k \log \frac{K \cdot \exp\left(\max_{1\leq j\leq K} x_j\right)}{\exp\left(-\max_{1\leq j\leq K} x_j\right)} \\
&= \frac{1}{\|\mathbf{x}\|_2} \left( \log K + \log \frac{\exp\left(\max_{1\leq j\leq K} x_j\right)}{\exp\left(-\max_{1\leq j\leq K} x_j\right)} \right) \\
&= \frac{1}{\|\mathbf{x}\|_2} \left( \log K + 2\max_{1\leq j\leq K} x_j \right) \\
&\leq \frac{\log K}{\|\mathbf{x}\|_2} + 2,
\end{aligned}
\tag{14}
$$

from which we can see that the magnitude of $\mathcal{L}_{CE}$ is bounded by a constant multiple of the 2-norm of $\mathbf{x}$. When $\|\mathbf{x}\|_2 \to +\infty$, although the exact form of how the norm of $\mathbf{x}$ is bounded by the model weights depend on many network details, *e.g.*, network architecture, activation functions.

---

**Algorithm 2** Ralif Framework including all selecting approaches.

---

**Input:** $L_0$: Initial labeled dataset, $U_0$: initial unlabeled dataset, $M$: model, $C$: AL cycles, $B$: query set, $P$: candidate set, $\phi_{M_{L_{c-1}}}(\cdot)$: convolutional encoder, and $\tau$: truncation hyperparameter.

1: **for** $c = 1, \ldots, C$ **do**
2:     Train model $M_{L_{c-1}}$ with labels set $L_{c-1}$
3:     $|P| = 0.1|U_{c-1}|$
4:     $\forall x_i \sim U_c$ generate pseudo-labels $\tilde{y}_i = \arg\max M_{L_{c-1}}(x_i)$
5:     **if** *single-IL* **then**
6:         **for** $x_s \in U_{c-1}$ **do**
7:             Generate pseudo-label $\tilde{y}_s = \arg\max M_{L_{c-1}}(x_s)$
8:             Calculate $\mathbb{E}_{(x_i \sim U_c, \tilde{y}_i \sim \tilde{p}_{Y|X})}[l(x_i, \tilde{y}_i; M_{L_c})]$ via (7)
9:         **end for**
10:    **end if**
11:    **if** *expectation-IL* **then**
12:        **for** $x_s \in U_{c-1}$ **do**
13:            Generate probabilistic predictions $\{p_k(x_s)\}_{k=1}^K$ from $M_{L_{c-1}}(x_s)$
14:            Calculate $\mathbb{E}_{(x_i \sim U_c, \tilde{y}_i \sim \tilde{p}_{Y|X})}[l(x_i, \tilde{y}_i; M_{L_c})]$ via (8)
15:        **end for**
16:    **end if**
17:    **if** *truncated-IL* **then**
18:        **for** $x_s \in U_{c-1}$ **do**
19:            Generate probabilistic predictions $\{p_k(x_s)\}_{k=1}^K$ from $M_{L_{c-1}}(x_s)$
20:            Calculate $\mathbb{E}_{(x_i \sim U_c, \tilde{y}_i \sim \tilde{p}_{Y|X})}[l(x_i, \tilde{y}_i; M_{L_c})]$ via (9)
21:        **end for**
22:    **end if**
23:    Select top-$|P|$ samples $\{x_i\}_{|P|}$ with minimum $\mathbb{E}_{(x_i \sim U_c, \tilde{y}_i \sim \tilde{p}_{Y|X})}[l(x_i, \tilde{y}_i; M_{L_c})]$
24:    Run *k-MEANS++* on $\{\phi_{M_{L_{c-1}}}(x_i)\}_{|P|}$ to select $|B|$ diverse samples
25:    $L_c = L_{c-1} \cup \{(x_s, y_s)\}_{s \in B}$
26: **end for**

**Return:** Final labeled dataset $L_C$ (of size $L_0 + C|B|$)

---

## A.2   Complete RALIF Algorithm

The complete Algorithm 2 including all three selecting approaches, *i.e.*, *single-IL* in (7), *expectation-IL* in (8) and *truncated-IL* in (9).

## A.3   Different thresholds

To investigate the impact of different values of $\tau$ on the performance of RALIF, we conducted experiments on a specific selection scenario. The experiments were performed using the Cifar10 dataset, a pretrained ResNet18 model architecture, and a budget size of 1000. We varied the value of $\tau$ across five different values: $\frac{0}{K} = 0$, $\frac{2}{K} = 0.2$, $\frac{4}{K} = 0.4$, $\frac{6}{K} = 0.6$, and $\frac{8}{K} = 0.8$. We then compared the resulting ACC and AUCA Ratio at the first and last selection cycle. The results are summarized in Table 3. It is interesting to note that regardless of the specific value of $\tau$, RALIF consistently outperforms the random baseline. Further, based on the results shown in Table 3, we observe that $\tau = \frac{4}{K}$ yields the best performance among the tested values.

## A.4   Diversity operation

Below we consider the same AL scenario as previously detailed in Section 5.1, *i.e.*, using the Cifar10 dataset, the pretrained ResNet18 model architecture, and a budget size of 1000. In the experiments, we replace the *k-MEANS++* algorithm in RALIF with the mean shift algorithm Fukunaga & Hostetler (1975). Notably, the mean shift algorithm circumvents the need for an explicit specification of the number of clusters. Thus, we first apply the mean shift algorithm to the selected candidate pool and then select the samples closest

Table 3: Comparison with different thresholds for the indicator function $\pi(\cdot)$ in (9). We show means and standard deviations of the results for all algorithms over three trials. The first row provides results for the random baseline, which serves as a reference for comparisons. The best results are highlighted in green.

| $\tau$ | Cycle | | AUCA Ratio |
| | First $(1^{st})$ | Last $(9^{th})$ | |
| --- | --- | --- | --- |
| Random | $85.31 \pm 0.29$ | $92.29 \pm 0.09$ | $1.000 \pm 0.000$ |
| 0.0 | $85.66 \pm 0.18$ | $92.92 \pm 0.16$ | $1.007 \pm 0.001$ |
| 0.2 | $86.60 \pm 0.10$ | $94.24 \pm 0.04$ | $1.019 \pm 0.002$ |
| 0.4 | $86.92 \pm 0.15$ | $94.35 \pm 0.05$ | $1.021 \pm 0.002$ |
| 0.6 | $86.59 \pm 0.16$ | $94.18 \pm 0.03$ | $1.019 \pm 0.002$ |
| 0.8 | $86.63 \pm 0.26$ | $94.03 \pm 0.05$ | $1.018 \pm 0.002$ |

Table 4: Diversity operations comparison results. We present the mean and standard deviation of the performance metrics for different diversity selection over three trials. Best results are highlighted in green.

| Candidate Selection | Diversity | | Cycle | | AUCA Ratio |
| | mean shift | k-MEANS++ | First $(1^{st})$ | Last $(9^{th})$ | |
| --- | --- | --- | --- | --- | --- |
| *Random* | | | $85.31 \pm 0.29$ | $92.29 \pm 0.09$ | $1.000 \pm 0.000$ |
| *truncated-IL* | ✓ | | $86.41 \pm 0.23$ | $94.16 \pm 0.16$ | $1.019 \pm 0.002$ |
| *truncated-IL* | | ✓ | $86.92 \pm 0.15$ | $94.35 \pm 0.05$ | $1.020 \pm 0.002$ |

to the centroid of each cluster, where the number of samples chosen from each cluster is determined by the proportion of samples in that cluster to the total number of candidate samples. Results presented in Table 4 demonstrate that both the mean shift algorithm and the *k-MEANS++* algorithm can enhance RALIF as a diversity operation. However, it is worth noting that the *k-MEANS++* algorithm yields better performance.

## A.5 Additional AL scenarios

In Table 5, we present the results of additional experiments conducted under different active learning scenarios relative to those in Table 2. In these experiments, we consider ResNet18 (scratch), identical to the one introduced in Sec 5, alongside two other model architectures: VGG16 Simonyan & Zisserman (2014) and EfficientNet-B0 Tan & Le (2019), both of which are initialized with pretrained weights from ImageNet Deng et al. (2009). Additionally, we use a different dataset, SVHN Netzer et al. (2011). For a more detailed analysis, we also provide a comprehensive view of the performance comparisons across all selection cycles and present boxplots summarizing the AUCA Ratio in Figure 3.

## A.6 Additional Comparison with Influence Selection for Active Learning (ISAL)

To compare the proposed RALIF with ISAL, as presented in Liu et al. (2021), we consider the active learning scenario described in Table 3 of Liu et al. (2021). In this scenario, the dataset is Cifar10, the initial labeled dataset size is 1000, and the active learning budget is set at 1000 samples. The model used here and the training details follow the *Target Model* specifications outlined in Liu et al. (2021). The results of the comparison between the proposed RALIF and ISAL are presented in Table 6. More details about the difference between ISAL, ISAL_v2 and ISAL_v3 can be found in *The Selection of Reference Set* Section in Liu et al. (2021). From the results, we see that in the same active learning scenario, the proposed RALIF performs better than ISAL, regardless of the selection of reference set used in ISAL.

## A.7 Significance Tests

To quantify the significance of the performance differences between RALIF and other approaches, we employ a two-sided permutation test (Greenwood, 2014). Specifically, we start with predictions for the test dataset

Table 5: Comparison with baselines in different scenarios. $B$: budget size at each selection cycle. Note that for RALIF we use *truncated-IL* for candidate selection and *K-MEANS++* for diversity down-selection. We present the mean and standard deviation of the results for all algorithms over three trials. The best and worse than *Random* results are highlighted in green and red, respectively. Underlined values indicate instances where the performance differences between RALIF and other approaches is not statistically significant, *i.e.*, the $p$-value for the significance test exceeds the 0.05 threshold.

| Datasets | Setting $B$ | models | Methods | Cycles First($1^{st}$) | Last ($9^{th}$) | AUCA Ratio |
|---|---|---|---|---|---|---|
| SVHN | 1000 | *ResNet18 (scratch)* | Random | $86.02 \pm 0.311$ | $93.74 \pm 0.263$ | $1.000 \pm 0.000$ |
| | | | Uncertainty | $85.47 \pm 2.049$ | $\underline{95.58 \pm 0.138}$ | $1.015 \pm 0.004$ |
| | | | Coreset | $86.29 \pm 1.245$ | $95.24 \pm 0.110$ | $1.014 \pm 0.003$ |
| | | | BADGE | $87.55 \pm 0.701$ | $95.38 \pm 0.148$ | $1.018 \pm 0.003$ |
| | | | Grad-Norm | $87.31 \pm 0.965$ | $95.55 \pm 0.172$ | $1.018 \pm 0.004$ |
| | | | RALIF | $88.14 \pm 0.679$ | $95.77 \pm 0.143$ | $1.023 \pm 0.003$ |
| Cifar10 | 1000 | *VGG16* | Random | $84.26 \pm 0.665$ | $91.33 \pm 0.248$ | $1.000 \pm 0.000$ |
| | | | Uncertainty | $85.22 \pm 0.335$ | $\underline{93.15 \pm 0.219}$ | $1.014 \pm 0.001$ |
| | | | Coreset | $82.58 \pm 1.016$ | $90.49 \pm 0.194$ | $0.984 \pm 0.001$ |
| | | | BADGE | $85.61 \pm 0.311$ | $\underline{93.15 \pm 0.143}$ | $1.016 \pm 0.002$ |
| | | | Grad-Norm | $85.19 \pm 0.120$ | $\underline{92.98 \pm 0.255}$ | $1.015 \pm 0.002$ |
| | | | RALIF | $86.16 \pm 0.162$ | $93.47 \pm 0.167$ | $1.020 \pm 0.002$ |
| Cifar10 | 1000 | *EfficientNet-B0* | Random | $90.06 \pm 0.238$ | $95.02 \pm 0.236$ | $1.000 \pm 0.000$ |
| | | | Uncertainty | $\underline{91.20 \pm 0.127}$ | $96.58 \pm 0.039$ | $1.015 \pm 0.001$ |
| | | | Coreset | $90.78 \pm 0.210$ | $96.18 \pm 0.101$ | $1.010 \pm 0.002$ |
| | | | BADGE | $\underline{91.23 \pm 0.074}$ | $96.56 \pm 0.014$ | $1.015 \pm 0.003$ |
| | | | Grad-Norm | $\underline{91.31 \pm 0.047}$ | $96.52 \pm 0.105$ | $1.015 \pm 0.002$ |
| | | | RALIF | $91.61 \pm 0.172$ | $96.88 \pm 0.019$ | $1.018 \pm 0.003$ |

Table 6: We compared RALIF and ISAL on the Cifar10 dataset, utilizing the results for ISAL obtained from *Table 3* in Liu et al. (2021). For RALIF, we conducted the experiments three times for robustness and reported the performance mean, following Liu et al. (2021).

| Method | 1000 | 3000 | 5000 | 7000 | 9000 |
|---|---|---|---|---|---|
| ISAL | 45.52 | 67.72 | 81.24 | 85.96 | 89.26 |
| ISAL_v2 | 45.52 | 67.06 | 80.57 | 85.71 | 88.92 |
| ISAL_v3 | 45.52 | 67.12 | 80.11 | 84.88 | 88.71 |
| RALIF | 45.25 | **74.62** | **82.77** | **87.32** | **89.89** |

generated by different approaches. The test statistic, denoted as $T_{obs}$, represents the accuracy or AUCA Ratio differences between RALIF and other approaches, calculated using true test dataset labels. To obtain permuted results, denoted as $T^*$, we perform random permutations of the true labels of the test dataset $10,000$ times. Each time, we calculate the accuracy or AUCA Ratio differences between RALIF and other approaches using the generated permuted test dataset labels. Subsequently, we define the $p$-value as the proportion of times the absolute value of $T^*$ is greater than or equal to the absolute value of $T_{obs}$. A smaller $p$-value indicates a reduced likelihood that the observed accuracy or AUCA Ratio differences between RALIF and other approaches are due to random chance. In each active learning scenario, we conducted permutation tests three times (one per trial) and to be conservative, we only report the largest $p$-value (worst case) for each scenario in Table 7. The $p$-values in Table 7 demonstrate that RALIF's performance is statistically superior to that of other methods, further emphasizing the uniqueness of RALIF as an active learning algorithm with a strong emphasis on reliability. Additionally, for a concise overview of the frequency with which RALIF's performance is statistically significant, *i.e.,* $p$-values are smaller than 0.05 (the significance level threshold), compared to other methods, we provide a summary in Table 2.

Table 7: We conducted permutation test three times (one per trial) for each setting and only report the largest $p$-value observed for each setting. To improve readability, we have highlighted cases where the difference between RALIF and other approaches is statistically significant in green, *i.e.*, where the $p$-value is less than the 0.05 threshold.

| Datasets | Setting $B$ | models | Methods | First($1^{st}$) | Last ($9^{th}$) | AUCA Ratio |
|---|---|---|---|---|---|---|
| Cifar10 | 1000 | ResNet18 (scratch) | Random | 0.000 | 0.000 | 0.000 |
| | | | Uncertainty | 0.000 | 0.858 | 0.965 |
| | | | Coreset | 0.000 | 0.000 | 0.000 |
| | | | BADGE | 0.000 | 0.003 | 0.022 |
| | | | Grad-Norm | 0.000 | 0.000 | 0.000 |
| Cifar10 | 1000 | ResNet18 | Random | 0.000 | 0.000 | 0.003 |
| | | | Uncertainty | 0.000 | 0.099 | 0.611 |
| | | | Coreset | 0.000 | 0.000 | 0.032 |
| | | | BADGE | 0.434 | 0.692 | 0.987 |
| | | | Grad-Norm | 0.664 | 0.756 | 0.645 |
| Cifar10 | 2000 | ResNet18 | Random | 0.000 | 0.000 | 0.001 |
| | | | Uncertainty | 0.695 | 0.000 | 0.667 |
| | | | Coreset | 0.000 | 0.000 | 0.000 |
| | | | BADGE | 1.000 | 0.783 | 0.843 |
| | | | Grad-Norm | 0.009 | 0.189 | 0.763 |
| Cifar100 | 1000 | ResNet18 | Random | 0.020 | 0.000 | 0.608 |
| | | | Uncertainty | 0.051 | 0.000 | 0.842 |
| | | | Coreset | 0.000 | 0.000 | 0.673 |
| | | | BADGE | 0.127 | 0.740 | 0.907 |
| | | | Grad-Norm | 0.948 | 0.000 | 0.838 |
| iNaturalist | 1000 | ResNet18 | Random | 0.000 | 0.000 | 0.000 |
| | | | Uncertainty | 0.000 | 0.000 | 0.000 |
| | | | Coreset | 0.000 | 0.000 | 0.000 |
| | | | BADGE | 0.000 | 0.001 | 1.000 |
| | | | Grad-Norm | 0.053 | 0.000 | 0.366 |
| iNaturalist | 100 | CLIP | Random | 0.000 | 0.000 | 0.000 |
| | | | Uncertainty | 0.000 | 0.000 | 0.000 |
| | | | Coreset | 0.000 | 0.000 | 0.000 |
| | | | BADGE | 0.000 | 0.216 | 1.000 |
| | | | Grad-Norm | 0.000 | 0.013 | 0.362 |
| SVHN | 100 | ResNet18 | Random | 0.000 | 0.000 | 0.000 |
| | | | Uncertainty | 0.000 | 0.924 | 0.050 |
| | | | Coreset | 0.000 | 0.000 | 0.011 |
| | | | BADGE | 0.178 | 0.000 | 0.271 |
| | | | Grad-Norm | 0.000 | 0.002 | 0.352 |
| Cifar10 | 100 | VGG16 | Random | 0.000 | 0.000 | 0.009 |
| | | | Uncertainty | 0.000 | 0.105 | 0.388 |
| | | | Coreset | 0.000 | 0.000 | 0.000 |
| | | | BADGE | 0.022 | 0.064 | 0.657 |
| | | | Grad-Norm | 0.000 | 0.000 | 0.673 |
| Cifar10 | 100 | EfficientNet-B0 | Random | 0.000 | 0.000 | 0.010 |
| | | | Uncertainty | 0.096 | 0.002 | 0.648 |
| | | | Coreset | 0.000 | 0.000 | 0.091 |
| | | | BADGE | 0.152 | 0.000 | 0.495 |
| | | | Grad-Norm | 0.328 | 0.001 | 0.537 |

## A.8   Detail results comparison

For a more detailed comparison between the baselines and RALIF, we present the ACC across all AL selection cycles and AUCA Ratio boxplots for all baselines and RALIF under all six AL scenarios in Figure 3.

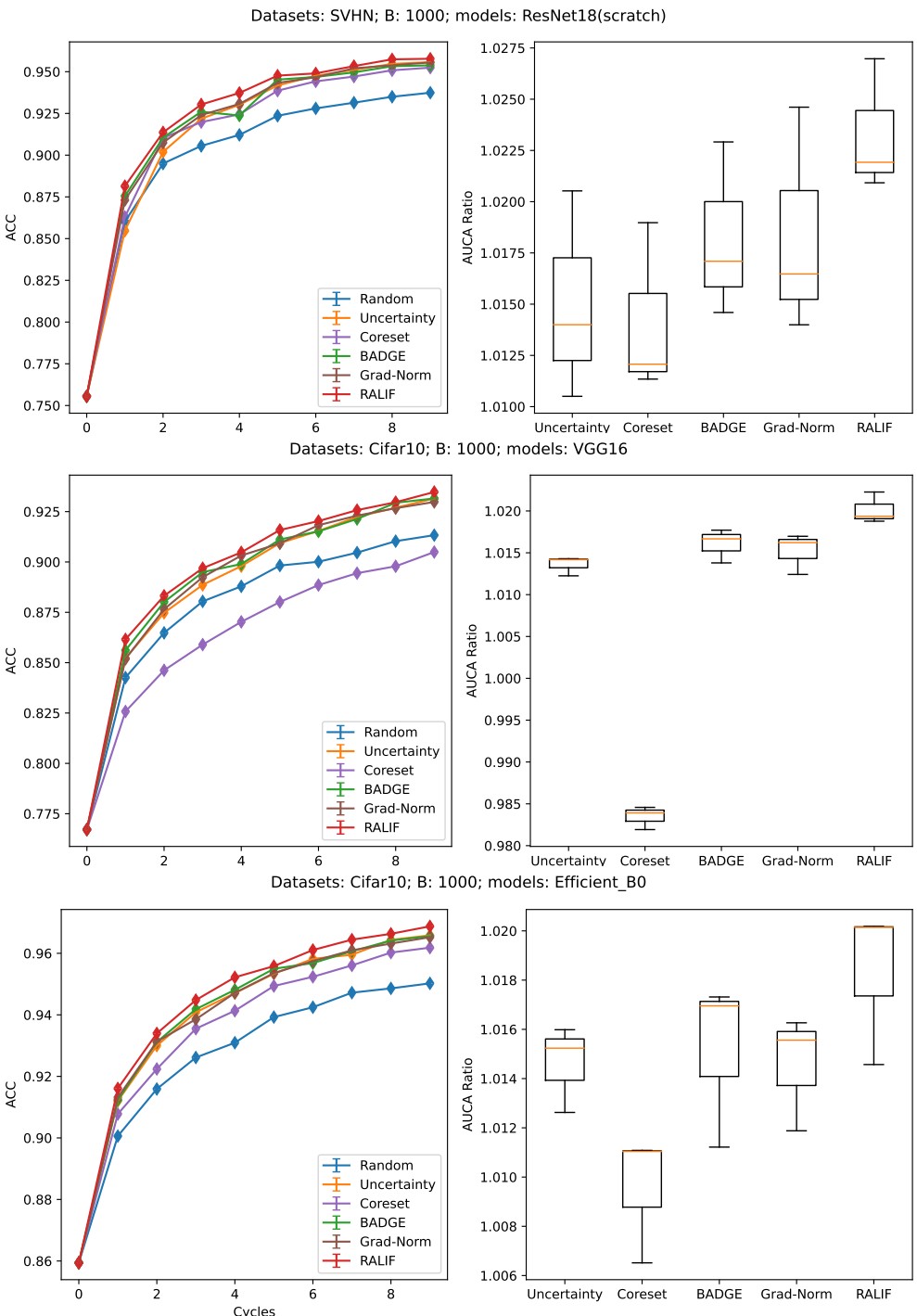

Figure 3:   Detailed AL results. Each row corresponds to an AL scenario, following the same order as in Table 2. The title for each row provides a description of the AL scenario, including the Dataset, B (budget size at each selection cycle), and the model used. Standard deviations are omitted in the ACC plots for clarity, and AUCA Ratio boxplots summarize results over the three trials.

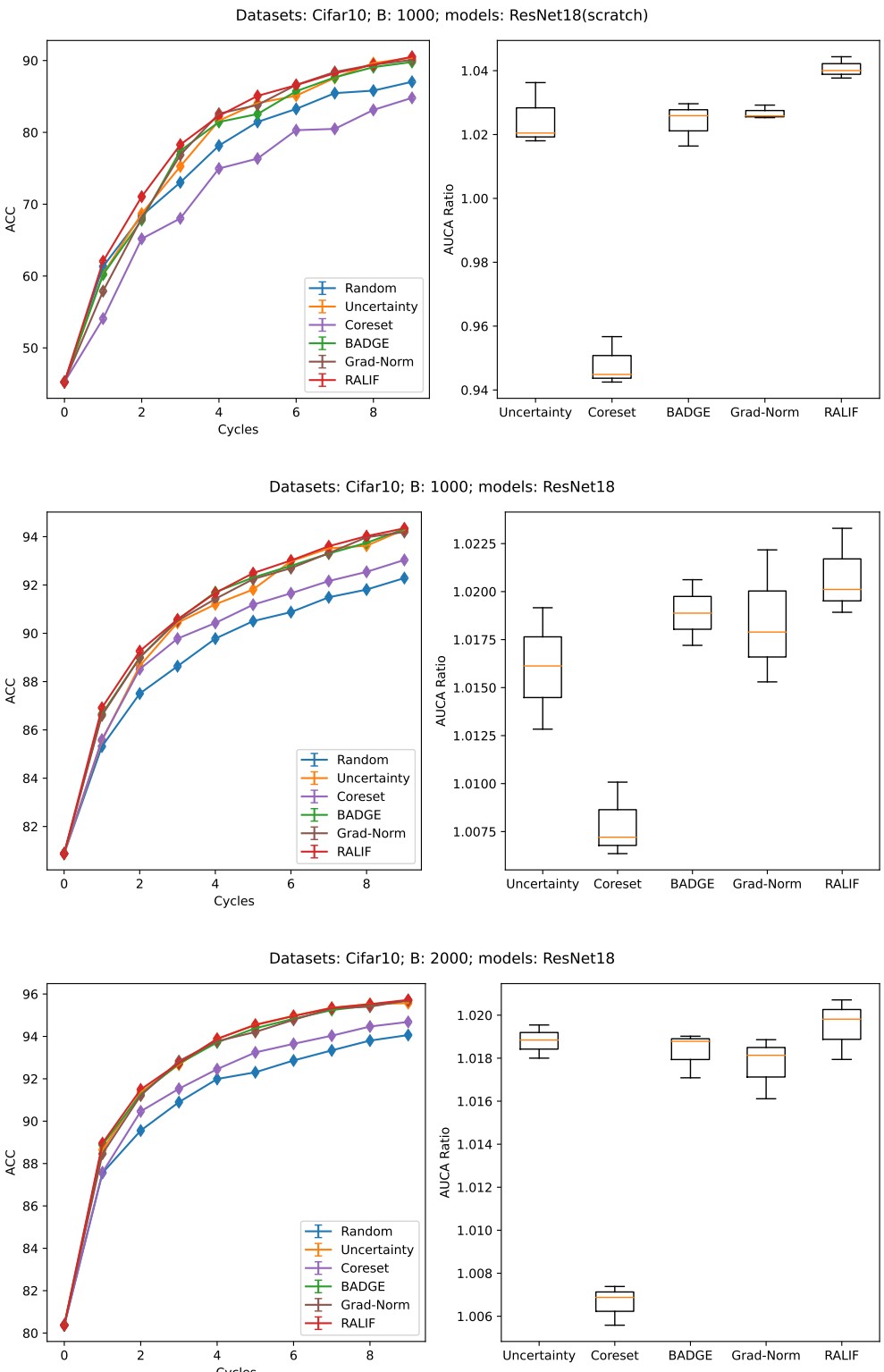

Figure 3: Detailed AL results. Each row corresponds to an AL scenario, following the same order as in Table 2. The title for each row provides a description of the AL scenario, including the Dataset, B (budget size at each selection cycle), and the model used. Standard deviations are omitted in the ACC plots for clarity, and AUCA Ratio boxplots summarize results over the three trials.

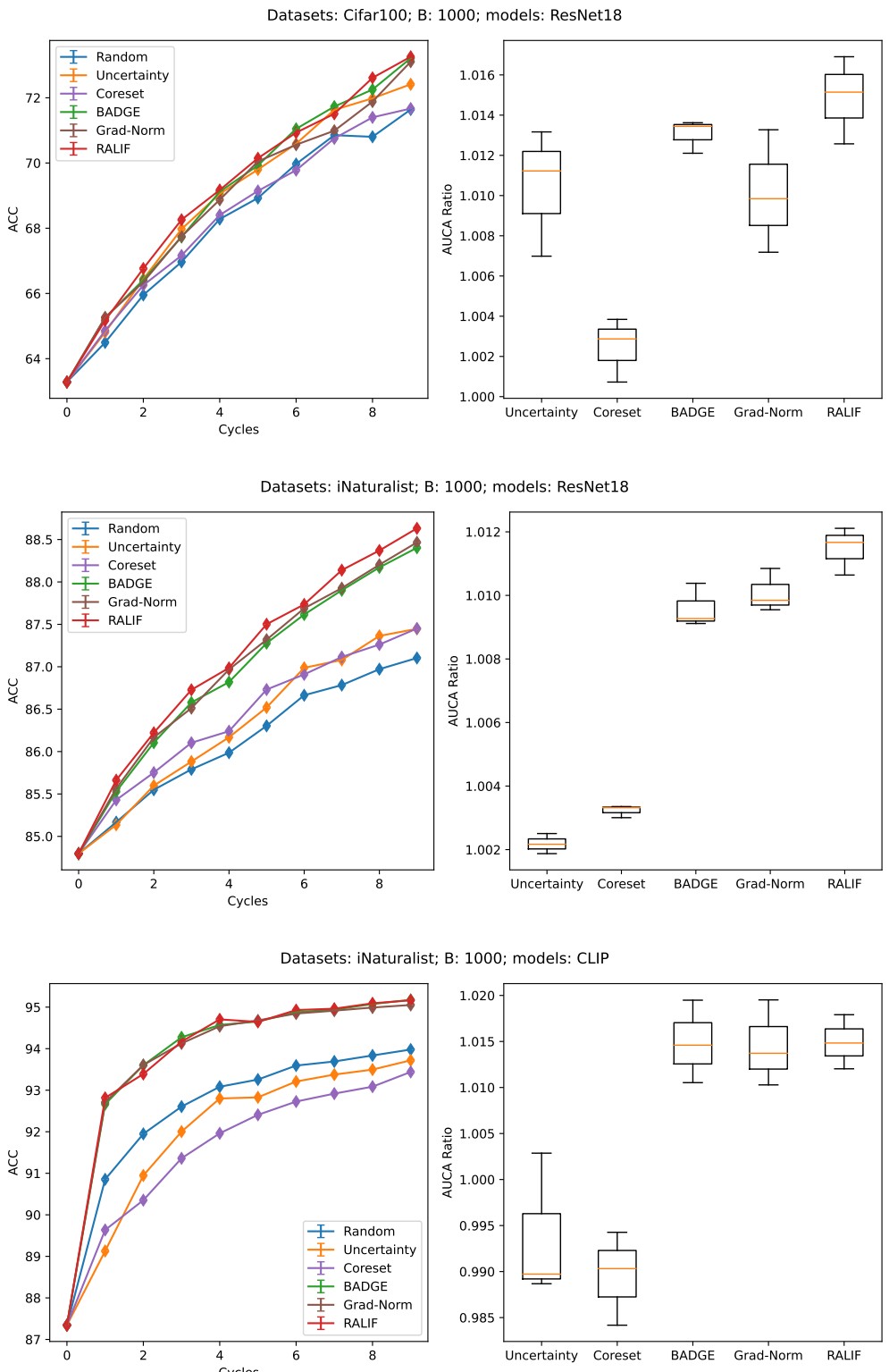

Figure 3: Detailed AL results. Each row corresponds to an AL scenario, following the same order as in Table 2. The title for each row provides a description of the AL scenario, including the Dataset, B (budget size at each selection cycle), and the model used. Standard deviations are omitted in the ACC plots for clarity, and AUCA Ratio boxplots summarize results over the three trials.

