# OpenReview forum: "Reliable Active Learning via Influence Functions"
_TMLR — Accepted by TMLR_

### Review · Reviewer_Qj5H · 2023-08-08

**Summary Of Contributions:**

1. The paper proposed a theoretical motivated method for active learning, i.e., using initial labeled data to pretrain a model for generating pseudo labels of unlabeled data; then select unlabeled data based on influence functions to acquire oracle labels.
2. The proposed method is evaluated on CIFAR-10, CIFAR-100 and iNaturalist and compared with representative baselines in active learning.

**Audience:**

Yes

**Claims And Evidence:**

Yes

**Requested Changes:**

1. Explain the similarity & difference between the proposed method and gradient based baselines, e.g., BADGE and Grad-Norm.
2. Add significance test for results in Table 2 and highlight the best and comparable methods.


**Strengths And Weaknesses:**

- Strengths:
1. The paper is well-written and presented. The related work, the proposed method and the empirical results are presented in a clear and logical way.
2. The proposed method is theoretical motivated and can be computed efficiently without retraining the model when selecting the unlabeled samples.

- Weaknesses:
1. The classification accuracy of the proposed method may not be very much advantageous when compared with baselines, e.g., BADGE and Grad-Norm, in most settings in Table 2. It seems that the proposed method is also based on the gradient information (i.e., influence function) which is quite similar with the baselines.
2. It may be hard to quickly check the performance comparison of methods from the first glance in Table 2 due to the lack of tests for significance.
3. The performance of the proposed method may be affected if the number of initial labeled data is too small to train a good enough model that could produce reliable pseudo-labels at the beginning of cycles.

---

> ### Author Response · Authors · 2023-10-04
> **The response for requested changes**
>
> We appreciate your valuable suggestions (all changes in the revision text are highlighted in red):
>
> 1. We have expanded our discussions regarding the distinctions and commonalities between our RALIF approach and other methods, which can now be found under the 'AL for deep neural networks' section within the 'Related work' section of our paper.
>
> 2. We have introduced significance test results and provided detailed information about it in Appendix A.7.

---

### Review · Reviewer_n6fZ · 2023-09-14

**Summary Of Contributions:**

This paper addresses the challenge of insufficient labeled data in deep learning models by introducing a theoretically motivated active learning framework called "Reliable Active Learning via Influence Functions" (RALIF). RALIF consistently outperforms random selection and existing active learning methods by efficiently identifying valuable data samples that improve model performance across the entire dataset.

**Audience:**

Yes

**Broader Impact Concerns:**

This paper provide a good pipeline for active learning. The following research can develop more effective estimation functions to improve this pipeline.

**Claims And Evidence:**

Yes

**Requested Changes:**

The detailed changes are listed on the previous block.

**Strengths And Weaknesses:**

Strengths:
1. This paper is theoretically solid and have improved performance.

Weaknesses:
1. limited experiments: most experiments are conduct on a relative small model: ResNet18. It would be better if the author try more models and datasets.
2. Since IL is the most important part of the algorithm, the paper lack discussion on IL.
3. It would be better if Figure 1 can add more text in the arrow lines to illustrate how the pipeline works.
4. The pipeline seems simple and effective. I would like to see more ablation study about ranking score like IL to see the difference.

---

> ### Author Response · Authors · 2023-10-04
> **The response for requested changes**
>
> We appreciate your support for the proposed pipeline and suggestions (all changes in the revision text are highlighted in red):
>
> 1. Given the extensive content in Table 2, we have incorporated additional experiments in Appendix A.6. We include two more models, namely VGG16 and EfficientNet_B0, along with the introduction of a new dataset, SVHN.
>
> 2. We added more details about the influence function in Section 4.2.
>
> 3. We have incorporated additional explanatory text, highlighted in red in Figure 1, to provide a clearer understanding of how our pipeline works.
>
> 4. We provided the ablation study results in Table 1, where we verify the impact of each component within the proposed pipeline. The second row in Table 1, denoted as ‘single-IL without Diversity k-MEANS++,’ corresponds to the results when only using IL as ranking score to select samples. By comparing these results with the results of Random selection, indicated in the first row of Table 1, we could see only using IL as ranking score can select valuable samples. Furthermore, by comparing the performance of the proposed truncated-IL algorithm, represented in the fourth row of Table 1, with the results of only using IL as a ranking score, we could see the proposed truncated-IL algorithm has a better performance.
> In addition, from the comparison between the second row and fifth row in Table 1, where the second row represents only using IL as ranking score and the fifth row reflects using IL as ranking score combined with proposed diversity operation, we could see our diversity operation effectively enhances the selection of valuable samples.
> Responding to the suggestions from reviewer Frwm, we have also expanded our ablation study in Table 4 to examine the impact one additional diversity operation as part of our RALIF.

---

> > ### Comment · Reviewer_n6fZ · 2023-10-22
> >
> > I have read the authors' response and feedback from other reviewers. My concerns have been solved.

---

### Review · Reviewer_Frwm · 2023-09-21

**Summary Of Contributions:**

This paper introduces a novel active learning framework, Reliable Active Learning via Influence Functions (RALIF) which utilizes influence functions, offering a theoretically sound selection mechanism. This mechanism enhances the likelihood of choosing samples that, when labeled and incorporated into the training dataset, will boost model performance.

**Audience:**

Yes

**Claims And Evidence:**

Yes

**Requested Changes:**

See Weaknesses

**Strengths And Weaknesses:**

Strengths
1. The paper is well-written and presented in a clear and understandable manner, making it easy for readers to follow the content.

2. The concept is presented in a straightforward manner with a sound derivation.


Weaknesses:

1. The enhancements, when compared to other methods, show only marginal gains. Refer to the average performance presented in Table 2.

2. Given the similarities between semi-supervised learning (SSL) and active learning, it would be pertinent to evaluate RALIF within a more sophisticated SSL framework. Potential SSL algorithms to consider are outlined in [1]. For RALIF to be deemed valuable, it should ideally outperform the SSL benchmarks presented in [1]; otherwise, its utility comes into question.

3. The study lacks an ablation analysis concerning the substitution of K-means++ with alternative clustering methods, such as mean shift or DBSCAN.

[1] USB: A Unified Semi-supervised Learning Benchmark for Classification.

---

> ### Author Response · Authors · 2023-10-04
> **The response for requested changes**
>
> We appreciate your valuable suggestions (all changes in the revision text are highlighted in red):
>
> 1. While we acknowledge that the metrics, such as accuracy and AUCA Ratio, presented in Table 2 may not reveal seemingly large improvements when compared to other state-of-the-art methods, it is essential to emphasize the primary objective of our work: the development of a dependable active learning algorithm with applicability in diverse scenarios. Although the results in Table 2 may not showcase such large enhancements, they consistently demonstrate that our proposed RALIF outperforms random selection. Importantly, we have included the results of significance tests in Appendix A.7 to illustrate the statistical significance of the performance differences between RALIF and other approaches. These tests provide robust evidence that RALIF performs significantly better than other approaches.
>
> 2. We appreciate your question. However, it is essential to note that the primary focus of active learning and semi-supervised learning (SSL) differs. Active learning is primarily concerned with annotating valuable samples from an unlabeled pool, whereas semi-supervised learning focuses on effectively utilizing the samples within the unlabeled pool. Consequently, these two tasks can be viewed as complementary to each other, stemming from their distinct motivations.
> In practical scenarios where annotation costs are constrained, we can think of a common strategy that involves initially applying an active learning algorithm to select valuable samples for annotation. Subsequently, semi-supervised learning algorithms are employed to further enhance the performance of the task model. To validate this assumption, we conducted an experiment: we employed the CIFAR-10 dataset and utilized a ResNet18 (scratch) as the task model. Given an annotation budget of 2000 samples, we initially randomly annotated 1000 samples. Following this, we ran our RALIF algorithm to select an additional 1000 samples for annotation, resulting in a total of 2000 annotated samples. We then executed the MixMatch algorithm using these 2000 annotated samples in conjunction with the remaining unlabeled samples. For comparative purposes, we also ran the MixMatch algorithm using the same initial 1000 samples, added to another set of randomly annotated 1000 samples, along with the remaining unlabeled samples. This experiment was repeated three times.
> The results of combining our RALIF with MixMatch yielded a performance of 93.797±0.216, while solely running MixMatch resulted in a performance of 93.547±0.532. This comparison shows that our RALIF can potentially improve the performance of MixMatch, affirming the synergy between active learning and semi-supervised learning (SSL) algorithms.
>
> 3. We have provided the ablation study about using mean shift as the diversity operation in Appendix A.5.

---

> ### Comment · Reviewer_Frwm · 2023-10-21
>
> I appreciate the author's response to my questions; my concerns have been adequately addressed.

---

### Review · Reviewer_RLtK · 2023-09-22

**Summary Of Contributions:**

This work introduces a theoretically motivated active learning framework that relies on estimating sample value through influence functions, pseudo-labels, and diversity selection. The authors provide comprehensive theoretical proofs and analyses for the proposed method, and they conducted extensive experiments to demonstrate its effectiveness.

**Audience:**

Yes

**Claims And Evidence:**

Yes

**Requested Changes:**

1. Add comparison against state of the arts.
2. Illustrate the difference of this work from other similar works.

**Strengths And Weaknesses:**

While this work introduces a theoretically motivated active learning framework based on estimating sample value through influence functions, pseudo-labels, and diversity selection, a significant issue in this paper is the absence of a discussion regarding the state-of-the-art active learning methods and how this work differs from them. The lack of a comparison against existing active learning methods leaves us uncertain about where this paper stands in the literature and its relative contributions.

The primary novelty of this paper revolves around the selection of the most valuable samples for labeling using influence functions. However, it's worth noting that similar approaches have been proposed in the past, like Liu, Zhuoming, Hao Ding, Huaping Zhong, Weijia Li, Jifeng Dai, and Conghui He. "Influence selection for active learning." In Proceedings of the IEEE/CVF International Conference on Computer Vision, pp. 9274-9283. 2021. What is the major difference between the proposed influence function and that of the referenced paper? It would be valuable for the authors to clarify the distinctions and provide an experimental comparison if applicable.

A bunch of literature is missing, what is the major difference between the propose method with those works?

Liu, Zhuoming, Hao Ding, Huaping Zhong, Weijia Li, Jifeng Dai, and Conghui He. "Influence selection for active learning." In Proceedings of the IEEE/CVF International Conference on Computer Vision, pp. 9274-9283. 2021.

Yi, John Seon Keun, Minseok Seo, Jongchan Park, and Dong-Geol Choi. "Pt4al: Using self-supervised pretext tasks for active learning." In European Conference on Computer Vision, pp. 596-612. Cham: Springer Nature Switzerland, 2022.

Vo, Huy V., Oriane Siméoni, Spyros Gidaris, Andrei Bursuc, Patrick Pérez, and Jean Ponce. "Active learning strategies for weakly-supervised object detection." In European Conference on Computer Vision, pp. 211-230. Cham: Springer Nature Switzerland, 2022.

Yoo, Donggeun, and In So Kweon. "Learning loss for active learning." In Proceedings of the IEEE/CVF conference on computer vision and pattern recognition, pp. 93-102. 2019.

Freytag, Alexander, Erik Rodner, and Joachim Denzler. "Selecting influential examples: Active learning with expected model output changes." In Computer Vision–ECCV 2014: 13th European Conference, Zurich, Switzerland, September 6-12, 2014, Proceedings, Part IV 13, pp. 562-577. Springer International Publishing, 2014.

No experimental comparison with the state of the arts.

---

> ### Author Response · Authors · 2023-10-04
> **The response for requested changes**
>
> We appreciate your valuable suggestions (all changes in the revision text are highlighted in red):
>
> 1. We acknowledge that our paper does not directly compare with the latest active learning algorithms. However, it is important to clarify that our paper primarily focuses on the proposal of a reliable active learning algorithm. Consequently, our emphasis lies in assessing whether our algorithm can consistently outperform random selection and remain competitive with other established active learning methods across various scenarios. The results in Table 2 clearly demonstrate that our RALIF consistently outperforms random selection, while other active learning algorithms do not consistently do so.
> Furthermore, the baseline algorithms selected in our paper represent a wide range of popular selection strategies used in the domain of active learning for deep learning. By comparing our method to these baselines, we can effectively illustrate that our algorithm's performance is indeed competitive with other popular active learning techniques. For instance, the inclusion of "Grad-Norm" as one of our baselines demonstrates its superior performance compared to the mentioned "Learning loss for active learning," indicating that our algorithm also perform better in comparison to the latter.
> Additionally, we have incorporated the results of a significance test in Appendix A.7, in response to the suggestions from reviewer Qj5H. These results affirm the statistical significance of our RALIF's superior performance over other approaches.
>
> 2. We extend our sincere gratitude to you for sharing those valuable related works, which we have included into both our related work section and introduction section.
> Of the mentioned similar works, "Influence selection for active learning" bears the closest relationship to our research. As such, we have dedicated a detailed discussion in our related works section to elucidate the distinctions between our RALIF approach and their work.
> For the other works, we have integrated them into our introduction sections and here, provided detail explanations of the key differences between our research and theirs:
>   In the case of 'Pt4al: Using self-supervised pretext tasks for active learning' and 'Learning loss for active learning.' It is important to note that the fundamental selecting rule behind these methods differs from our approach. Both 'Pt4al' and 'Learning loss' aim to select samples with high task loss, indicating that these samples are more informative for the current model. In contrast, our RALIF places its focus on selecting samples that have the potential to reduce the task loss for the entire dataset.
> In the case of “Active learning strategies for weakly-supervised object detection”, we appreciate your reference to a related active learning algorithm for object detection. However, our current RALIF is specifically designed for the selection of valuable samples in classification tasks. While we may consider extending our algorithm to object detection tasks in the future, such an expansion would serve as a powerful baseline for our research.
> In the case of “Selecting influential examples: Active learning with expected model output changes”, It's worth noting that the selection criterion employed in this paper differs from our approach. While this paper focuses on selecting samples that induce the most significant changes in model outputs, our RALIF prioritizes the selection of samples that can effectively reduce the overall task loss for the entire dataset. Notably, the authors of the referenced paper mention in their work that the change in model output can be considered an upper bound for the reduction in task loss. This suggests that in theory, their selection metric serves as an upper limit for our selection metric. Furthermore, it's important to highlight that this paper is tailored for Gaussian process regression, whereas our algorithm is specifically designed for deep learning models.

---

### Decision · Action_Editor_pdev · 2023-11-06

**Recommendation:** Accept with minor revision

**Comment:**

A 'minor revision' for this paper is recommended due to several remaining concerns. While the proposed RALIF method consistently shows improvements, the reviewers found these gains to be relatively marginal. Additionally, the choice of baselines may not be the latest. The authors included a discussion with the related work "Influence Selection for Active Learning" by Liu et al. (2021) in the paper, but there is a lack of further in-depth empirical discussion and analysis. It's important to note that Liu et al. (2021) clarified in their paper that they "chose the validation set V created in the first step of active learning as the reference set, since this would not cause additional annotation," and they also included relevant studies in their experiments. The authors thus need to further clarify the distinction between the author's new work and existing works (Liu et al. 2021) and may consider including relevant empirical comparisons or discussion to emphasize the advantages of the proposed new algorithm.

**Audience:**

The paper presents a new active learning framework for deep learning architectures, RALIF, which has shown consistent improvements and the potential to enhance semi-supervised performance. This may be of interest to researchers and practitioners working in the fields of machine learning, deep learning, and active learning.

**Claims And Evidence:**

The submission presents evidence that the proposed RALIF method consistently demonstrates improvements and enhances semi-supervised performance, supported by the reviewers.

---

> ### Author Response · Authors · 2023-11-14
> **The response for action editor suggestions**
>
> We appreciate your support for the proposed method. We understand that Liu et al. (2021) clarified in their Section “3.2 The Influence of an Untrained Sample” that “In practice, we select the validation set V created in the first step of active learning as reference set, since this would not cause additional annotation.” However, it's worth noting that in their “Implementation Details” paragraph in Section “4. Experiment,” they mentioned “For all datasets, we use all parameters in ResNet-18 to calculate the influence, and we use the test set as reference set.” Therefore, our interpretation is that they do use the test set as the reference set in their practical experiments. Nevertheless, in case we misunderstood their approach, we have updated the paragraph about Liu et al. (2021) in our “Related Work” section. Furthermore, we conducted a comparative experiment between the proposed RALIF method and their ISAL algorithm in Appendix A.8. Specifically, we followed the exact same active learning scenario used by Liu et al. (2021) in their “Table 3”. The results consistently demonstrate that the proposed RALIF outperforms their method in this specific scenario.
>
> Based on the editor's suggestions, we have revised our paper and uploaded the camera ready version.